# Using aquatic animals as partners to increase yield and maintain soil nitrogen in the paddy ecosystems

Liang Guo[1†], Lufeng Zhao[1†], Junlong Ye[1], Zijun Ji[1], Jian-Jun Tang[1], Keyu Bai[2], Sijun Zheng[2,3], Liangliang Hu[1]*, Xin Chen[1]*

[1]College of Life Sciences, Zhejiang University, Hangzhou, China; [2]Bioversity International, Maccarese, Italy; [3]Yunnan Academy of Agricultural Sciences, Kunming, China

**Abstract** Whether species coculture can overcome the shortcomings of crop monoculture requires additional study. Here, we show how aquatic animals (i.e. carp, crabs, and softshell turtles) benefit paddy ecosystems when cocultured with rice. Three separate field experiments and three separate mesocosm experiments were conducted. Each experiment included a rice monoculture (RM) treatment and a rice-aquatic animal (RA) coculture treatment; RA included feed addition for aquatic animals. In the field experiments, rice yield was higher with RA than with RM, and RA also produced aquatic animal yields that averaged 0.52–2.57 t ha$^{-1}$. Compared to their corresponding RMs, the three RAs had significantly higher apparent nitrogen (N)-use efficiency and lower weed infestation, while soil N contents were stable over time. Dietary reconstruction analysis based on $^{13}$C and $^{15}$N showed that 16.0–50.2% of aquatic animal foods were from naturally occurring organisms in the rice fields. Stable-isotope-labeling ($^{13}$C) in the field experiments indicated that the organic matter decomposition rate was greater with RA than with RM. Isotope $^{15}$N labeling in the mesocosm experiments indicated that rice used 13.0–35.1% of the aquatic animal feed-N. All these results suggest that rice-aquatic animal coculture increases food production, increases N-use efficiency, and maintains soil N content by reducing weeds and promoting decomposition and complementary N use. Our study supports the view that adding species to monocultures may enhance agroecosystem functions.

*For correspondence:
zjuhull@126.com (LH);
chen-tang@zju.edu.cn (XC)

†These authors contributed equally to this work

Competing interest: The authors declare that no competing interests exist.

## Editor's evaluation

This is a well conducted experimental study in which rice monocultures are compared with rice-aquatic animal co-cultures at multiple sites over multiple years. Co-culture increases plant yield and adds animal yield, but requires extra input of animal feed. Overall co-culture benefits yield and sustainability.

## Introduction

Biological simplification and reliance on chemicals have increased the concern with the low levels of biodiversity in modern, intensive agriculture (*Li et al., 2007*; *Smith et al., 2008*; *Kremen et al., 2012*; *Ren et al., 2014*; *Brooker et al., 2021*). In natural ecosystems, experiments have shown that increases in species number can increase ecosystem productivity and stability (*Hector et al., 1999*; *Tilman et al., 2006*; *Loreau and de Mazancourt, 2013*; *Tilman et al., 2014*; *van der Plas, 2019*). These positive effects of species diversity on ecosystem functioning are mainly explained by niche

**eLife digest** Monoculture, where only one type of crop is grown to the exclusion of any other organism, is a pillar of modern agriculture. Yet this narrow focus disregards how complex inter-species interactions can increase crop yield and biodiversity while decreasing the need for fertilizers or pesticides. For example, many farmers across Asia introduce carps, crabs, turtles or other freshwater grazers into their rice paddies. This coculture approach yields promising results but remains poorly understood. In particular, it is unclear how these animals' behaviours and biological processes benefit the ecosystem.

To examine these questions, Guo, Zhao et al. conducted three separate four-year field experiments; they compared rice plots inhabited by either carp, mitten crabs or Chinese softshell turtles with fields where these organisms were not present.

With animals, the rice paddies had less weeds, better crop yields and steady levels of nitrogen (a natural fertiliser) in their soil. These ecosystems could breakdown organic matter faster, use it better and had a reduced need for added fertilizer. While animal feed was provided in the areas that were studied, carp, crabs and turtles obtained up to half their food from the field itself, eating weeds, algae and pests and therefore reducing competition for the crops.

This work helps to understand the importance of species interactions, showing that diversifying monocultures may boost yields and make agriculture more sustainable.

partitioning, facilitation, and complementary resource use (*Cardinale et al., 2002*; *Isbell et al., 2009*; *Cardinale, 2011*; *Brooker et al., 2021*).

Interspecific facilitation (which occurs when one species makes conditions more favorable for another species) or complementary resource use are common in terrestrial, marine, and wetland ecosystems (*Bruno et al., 2003*; *Brooker et al., 2007*; *He et al., 2013*; *Bulleri et al., 2015*; *Wright et al., 2017*). Plants can make the local environment more favorable for their co-existing partners by reducing thermal, drought, and salt stress (*Gómez-Aparicio et al., 2004*; *Gómez-Aparicio et al., 2008*; *Pretzsch et al., 2013*; *Anthelme et al., 2014*); by increasing nutrient availability *Li et al., 2007*; by removing competitors or deterring predators (*Callaway et al., 2005*; *Gómez-Aparicio et al., 2008*; *Flory et al., 2014*); and by stimulating beneficial soil microorganisms (*Hortal et al., 2013*; *Rodríguez-Echeverría et al., 2015*). Animals can also enhance plant growth and population development by improving the soil environment (*Daleo et al., 2007*; *Booth et al., 2019*; by removing competitors *Cushman et al., 2011*); or by facilitating dispersal of fruits and seeds (*Bronstein et al., 2006*; *Carlo et al., 2014*).

Facilitative interactions have recently been successfully applied to forest restoration in arid areas (*Gómez-Aparicio et al., 2004*); to the establishment of plant communities in salty marshes or on beaches (*Bruno, 2000*); and to coral reef restoration (*Abelson, 2006*). Researchers have also proposed that facilitation or resource complementarity between species may increase the sustainability of agricultural production (*Ren et al., 2014*; *Brooker et al., 2021*). Although an increase in species richness in a natural plant community is expected to result in an increase in plant mass, an increase in species richness in agriculture may not always lead to increases in yield due to competition for light or nutrients (*Omer et al., 2007*). Understanding how species may or may not benefit is therefore critical for using species diversity in agriculture.

Intercropping systems (e.g. the interplanting of corn or wheat with a legume) or the planting of cover crops are examples of the successful use of crop diversity in agriculture. In legume-based cropping systems, legume crops provide intercropped non-legume crops with symbiotically fixed nitrogen (N) (*Li et al., 2007*; *Tsialtas et al., 2018*) and with increased phosphorus (P) availability due to the lowering of soil pH by $N_2$-fixing bacteria (*Li et al., 2007*); these effects increase the yield of the intercropped species (*Li et al., 2007*). Other intercropping systems can increase the diversity of soil microorganisms, natural enemies, and pollinators (*Cardinale et al., 2003*; *Kremen et al., 2007*; *Letourneau et al., 2011*; *Norris et al., 2018*). Using diverse cover crops can also help reduce soil erosion and greenhouse gas emissions (*Kaye and Quemada, 2017*).

Because paddy fields provide a shallow water habitat suitable for some aquatic animals (e.g. carp, crabs, and softshell turtles), the coculturing of rice with aquatic animals has been practiced in many

countries (e.g. Bangladesh, China, Egypt, India, Indonesia, Myanmar, Malaysia, the Philippines, Thailand, and Vietnam) (*Halwart and Gupta, 2004*; *Frei and Becker, 2005b*; *Ahmed and Garnett, 2011*). Several rice-aquatic animal coculture systems (e.g. rice-carp, rice-crab, and rice-turtle) have been developed (*Hu et al., 2016*). Field surveys and experiments have shown that these coculture systems can increase rice yields and soil fertility while reducing the need for fertilizers and pesticides compared to rice monoculture (*Xie et al., 2011*; *Hu et al., 2016*; *Zhang et al., 2016*; *Guo et al., 2020*). Why coculturing these aquatic animals with rice can reduce the application of fertilizers and pesticides, however, is poorly understood. Understanding how aquatic animals contribute to the reductions in fertilizer and pesticide application in coculture systems would help the development of sustainable rice production.

Animal behaviors (e.g. moving and grazing) are important drivers of ecosystem processes (e.g. carbon and nutrient cycling, and energy flux) (*Vanni et al., 2006*; *Schmitz et al., 2018*; *McInturf et al., 2019*). In wetland or aquatic ecosystems, grazing, 'muddying', and burrowing by aquatic animals have important roles in nutrient cycling (*Vanni, 2002*; *Thrush et al., 2006*; *Devlin et al., 2015*; *Atkinson et al., 2021*). For paddy ecosystems in which rice and aquatic animals coexist, understanding whether and how the behavior of aquatic animals affects ecosystem processes and functions could help researchers predict the effects of a coculture system of rice and an aquatic animal, and could also help improve the coculture system.

In this study, we conducted three field experiments and three mesocosm experiments to determine how coculture with aquatic animals benefits a rice paddy ecosystem in terms of productivity, nutrient-use efficiency, and the stability of soil N content. Because some fresh water animals (e.g. carp, crabs, crayfish, and softshell turtles) that are cultured in fish ponds or in paddy fields are omnivores and may use weeds, algae, and phytoplankton as food, we expected that the coculture of these aquatic animals would increase rice yield by reducing competitors of rice. We also expected that the cocultured aquatic animals would promote organic matter decomposition because of their feeding activity and would thereby promote nutrient recycling in the paddy ecosystem. Feed is often applied in the form of pellets to increase the growth of aquatic animals in coculture systems (*Hu et al., 2016*; *Guo et al., 2020*), and significant percentages of the N in the feed is often unconsumed and unassimilated by the animals. We therefore expected that this unconsumed and unassimilated feed-N could be used by rice plants, resulting in higher N-use efficiency and a more stable soil N content in a coculture system than in a rice monoculture.

## Results

### Yield, soil N content, and N-use efficiency in the field experiments

We conducted three 4-year-long field experiments: one with rice-carp, one with rice-crabs, and one with rice-turtles. We found that rice yield was significantly higher in the RA treatment (the treatment with the coculture of rice and an aquatic animal) than in the RM treatment (the treatment with rice monoculture) in the rice-carp experiment ($F_{1,10}$=7.828, p = 0.019), the rice-crab experiment ($F_{1,10}$=5.957, p = 0.035), and the rice-turtle experiment ($F_{1,10}$=12.472, p = 0.005) (*Figure 1a*). Compared to the corresponding monoculture, average rice yield over the 4 years in the RA treatment was 9.13% ± 3.11% higher for rice-carp, 12.05% ± 1.16 higher for rice-crabs, and 8.69% ± 1.74 higher for rice-turtles. During the experimental period, the average annual aquatic animal yield (in t ha$^{-1}$) was 0.85 for rice-carp, 0.56 for rice-crab, and 2.66 for rice-turtle systems (*Figure 1a*).

Averaged across all 4 years, total soil N content was not significantly different in the RA vs. the RM treatment in all three experiments ($F_{1,10}$=0.294 and p = 0.687 for the rice-carp experiment; $F_{1,10}$=1.325 and p = 0.154 for the rice-crab experiment; and $F_{1,10}$=0.236 and p = 0.345 for the rice-turtle experiment) (*Figure 1b*). At the end of the experiments, total soil N contents had not changed relative to initial values in the RA treatment of the rice-carp ($t_5$ = −0.533, p = 0.631), rice-crab ($t_5$ = 0.213, p = 0.842), and rice-turtle systems ($t_5$ = −1.279, p = 0.259) (*Appendix 1—figure 1*).

Compared to the RM treatment, the RA treatment received extra N from fish feed (*Appendix 2—table 1*). Data from the 4 years of the experiments showed that apparent N-use efficiency (ANUE) was higher in the RA treatment than in the RM treatment for the rice-crab system ($F_{1,10}$=9.557, p = 0.011) and the rice-turtle system ($F_{1,10}$=7.302, p = 0.022) but not for the rice-carp system ($F_{1,10}$=0.209, p = 0.657) (*Figure 1c*).

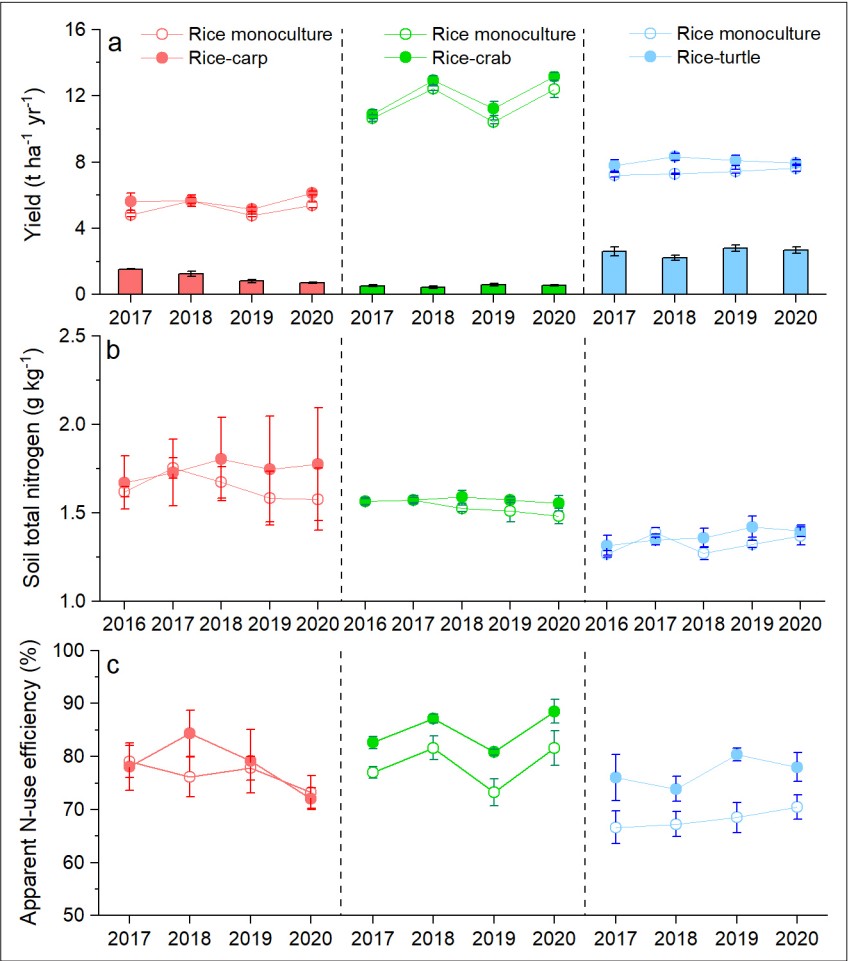

**Figure 1.** Yields of rice and aquatic animals (**a**), soil nitrogen content (**b**), and apparent N-use efficiency (**c**) in the field experiments. In (**a**), rice yields are indicated by symbols and lines, and aquatic animal yields are indicated by bars. Values are means ± SE (n = 6).

The online version of this article includes the following source data for figure 1:

**Source data 1.** Related to data in **Figure 1a–c**.

## Weed biomass, food sources, and decomposition in the field experiments

Weed biomass was significantly lower in the RA treatment than in the RM treatment in the rice-carp experiment ($F_{1,10}$=513.456, p = 0.000), the rice-crab experiment ($F_{1,10}$=538.032, p = 0.000), and the rice-turtle experiment ($F_{1,10}$=557.659, p = 0.000) (**Figure 2**). In all three experiments, weed biomass significantly decreased over time in the RA treatment (p < 0.05) but not in the RM treatment (p > 0.05).

Food source analysis showed that 50.2%, 34.9%, and 16.0% of the carp, crab, and turtle foods, respectively, were from the field environment rather than from applied feed (**Figure 3**). The main non-feed food sources for the aquatic animals in the rice fields included weeds, macro-algae, phytoplankton, zooplankton, and zoobenthos (**Figure 3**).

Determination of the stable isotope ($^{13}$C) content in maize leaves indicated that the percentage remaining in maize litter tubes at 40 days after the beginning (DAB) of the experiment was lower in the RA treatment than in the RM treatment in the rice-turtle experiment ($F_{1,10}$ = 23.353, p = 0.001) (**Figure 4**) but did not significantly differ between RM and RA treatments in the rice-carp experiment ($F_{1,10}$ = 0.076, p = 0.788) or the rice-crab experiment ($F_{1,10}$ = 1.092, p = 0.321) (**Figure 4**). At 80 DAB, however, the decomposition rate was higher in the RA treatment than in the RM treatment in all three experiments (for rice-carp: $F_{1,10}$ = 11.432, p = 0.007; for rice-crab: $F_{1,10}$=15.572, p = 0.003; for rice-turtle: $F_{1,10}$ = 14.349, p = 0.004) (**Figure 4**).

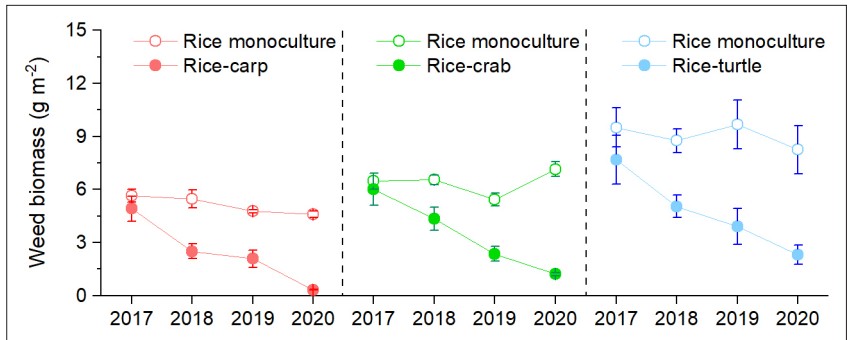

**Figure 2.** Weed biomass in the field experiments. No herbicides were used in the experiment. Values are means ± SE (n = 6).

The online version of this article includes the following source data for figure 2:

**Source data 1.** Related to data in *Figure 2*.

## Complementary utilization of feed-N by aquatic animals and rice in the mesocosm experiments

The δ[15]N percentage in the rice plant biomass was significantly higher in the RA treatment than in the RM treatment in all three mesocosm experiments (for rice-carp: $F_{1,10}$ = 1278, p = 0.000; for rice-crab: $F_{1,10}$ = 210.320, p = 0.000; for rice-turtle: $F_{1,10}$ = 91.572, p = 0.000) (*Appendix 1—figure 2*). The results from the mesocosm experiments also indicated that rice used from 13.02% to 35.13% of the feed-[15]N

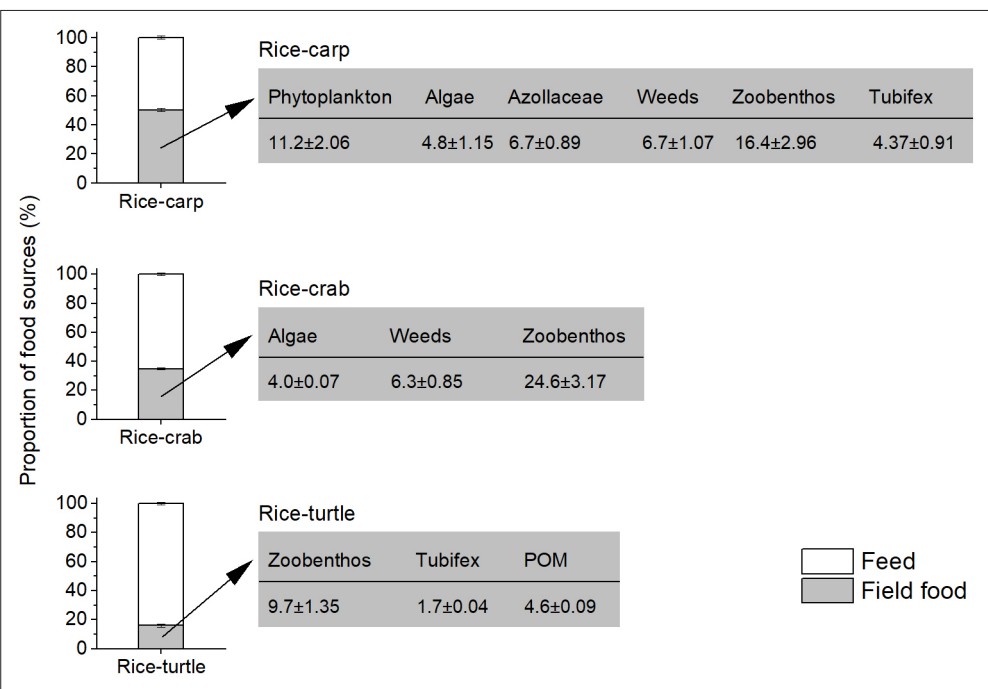

**Figure 3.** Diet components of aquatic animals as determined by dual stable isotopes (δ[13]C and δ[15]N), and the contribution of food sources to the aquatic animal diet in the field experiments. In each of the three plots in the figure, the white zone represents the proportion of food that aquatic animals (i.e. carp, crabs, or turtles) obtained from feed, and the grey zone represents the percentage of food that aquatic animals obtained from the rice field. The values in the rectangles to the right indicate the rice field food components as percentages of the total food obtained by the aquatic animals. POM: particulate organic matter.

The online version of this article includes the following source data for figure 3:

**Source data 1.** Related to data in *Figure 3*.

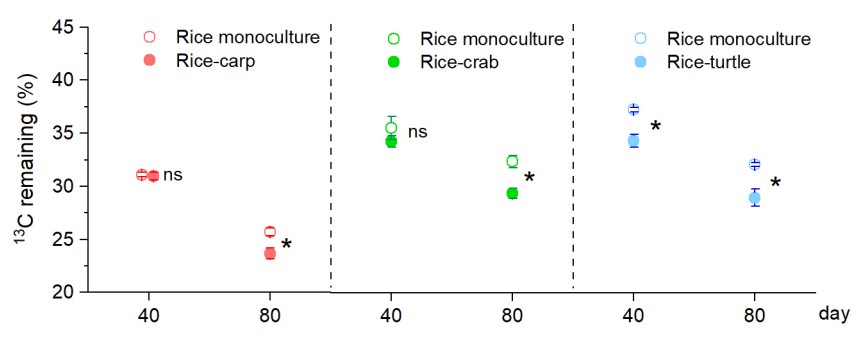

**Figure 4.** Organic matter decomposition in the field experiments at 40 and 80 days after the beginning (DAB) of the experiment. A higher percentage of $^{13}C$ remaining indicates slower decomposition. Values are means ± SE (n = 6). An asterisk indicates a significant difference between RM (rice monoculture) and RA (rice-aquatic animal coculture) at p < 0.05; ns indicates that the difference was not statistically significant.

The online version of this article includes the following source data for figure 4:

**Source data 1.** Related to data in *Figure 4*.

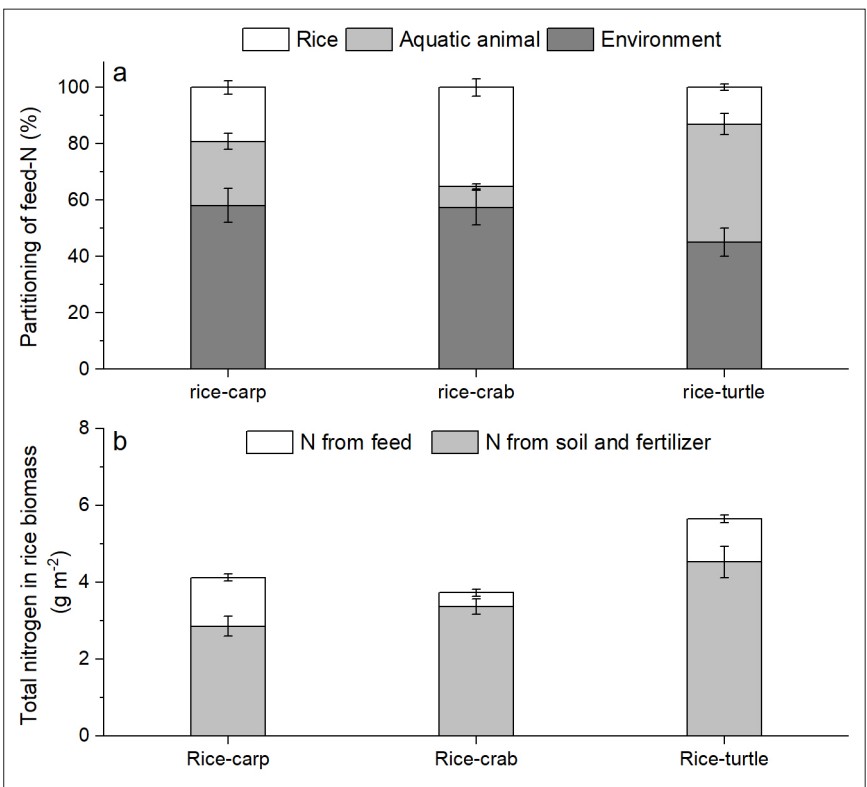

**Figure 5.** The fate of feed-N as determined by $^{15}N$ labeling in the mesocosm experiments. (**a**) Percentages of feed-N in rice plants, aquatic animals, and the environment (e.g., soil and water). (**b**) Total N in rice biomass at the end of the experiments. Values are means ± SE (n = 6).

The online version of this article includes the following source data for figure 5:

**Source data 1.** Related to data in *Figure 5a and b*.

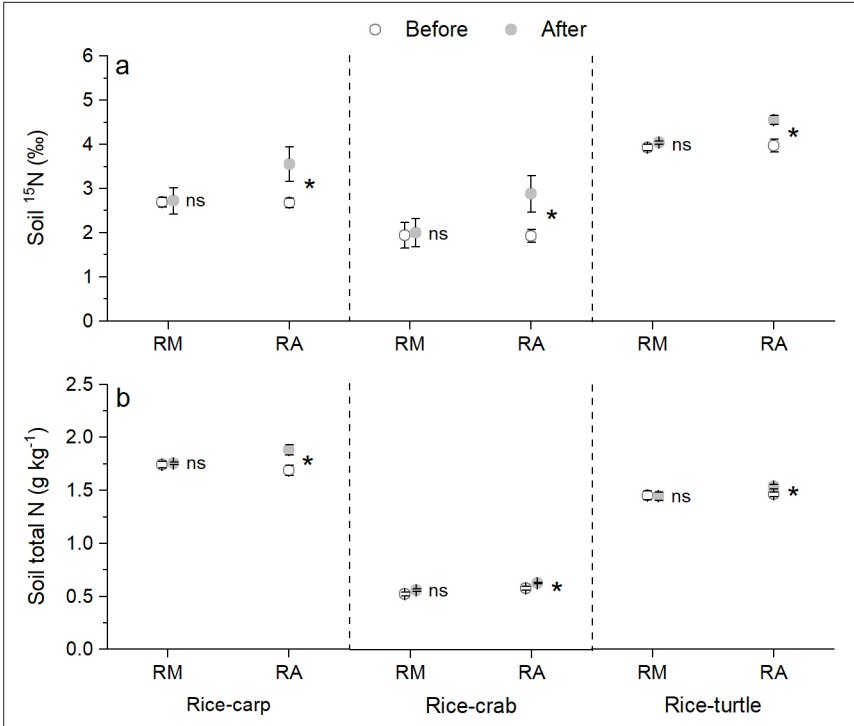

**Figure 6.** Soil δ15N content and total soil N content at the beginning vs the end of the mesocosm experiments. (**a**) δ15N value in soil at the beginning and end of the mesocosm experiments. (**b**) Total N in soil at the beginning and end of the mesocosm experiments. Values are means ± SE (n = 6). An asterisk indicates a significant difference between the before and after values for each treatment in each rice-aquatic system at p < 0.05; ns indicates that the difference was not statistically significant.

The online version of this article includes the following source data for figure 6:

**Source data 1.** Related to data in *Figure 6a and b*.

(*Figure 5a*). The N in feed that was not consumed by aquatic animals represented 9.61–30.65% of the rice biomass-N in the RA treatments (*Figure 5b*).

## N accumulation in the mesocosm experiments

The δ15N content in the soil in the mesocosm experiments did not significantly differ at the beginning vs. the end for the RM treatments (for rice-carp: $t_{10} = 0.131$, p = 0.449, n = 6; for rice-crab: $t_{10} = 0.115$, p = 0. 455, n = 6; for rice-turtle: $t_{10} = 0.523$, p = 0.623, n = 6), but was significantly higher at the end than at the beginning for the RA treatments (for rice-carp: $t_{10} = 2.178$, p = 0.027, n = 6; for rice-crab: $t_{10} = 2.153$, p = 0.028, n = 6; for rice-turtle: $t_{10} = 3.292$, p = 0.004, n = 6) (*Figure 6a*). The total N concentration in the soil was also significantly higher at the end than at the beginning of the experiments for the RA treatments (for rice-carp: $t_{10} = 2.765$, p = 0.009, n = 6; for rice crab: $t_{10} = 3.204$, p = 0.005, n = 6; for rice-turtle: $t_{10} = 2.519$, p = 0.015, n = 6) but not for the RM treatments (for rice-carp: $t_{10} = 0.477$, p = 0.322, n = 6; for rice-crab: $t_{10} = 1.774$, p = 0.053, n = 6; for rice-turtle: $t_{10} = 0.132$, p = 0.449, n = 6) (*Figure 6b*).

## Discussion

Researchers have been investigating whether the use of species interactions can overcome the limitations of monocultures, which depend on high fertilizer and pesticide input and which fail to take advantage of the possible beneficial effects of species interactions (*Kremen et al., 2012*; *Ren et al., 2014*; *Brooker et al., 2021*). Intercropping is an important and successful way to use biodiversity in agriculture (*Li et al., 2007*; *Brooker et al., 2015*; *Brooker et al., 2021*). The current study provides another example of exploiting positive species interactions in agriculture. In this case, the interactions

involve the use of an aquatic animal (i.e. carp, crabs, or turtles) as a partner crop for rice with the goals of stabilizing soil N and increasing the productivity and N-use efficiency of paddy ecosystems.

The role of animals in ecosystems has been increasingly recognized (*Vanni, 2002*; *Dirzo et al., 2014*; *Schmitz et al., 2018*). Animals can mediate carbon exchange between ecosystems, can mediate organic matter transformation within ecosystems, and can also drive nutrient cycling (*Schmitz et al., 2013*; *Schmitz et al., 2018*; *Vanni, 2002*; *McInturf et al., 2019*). In many aquatic ecosystems, aquatic animals can also significantly affect the plant community, primary productivity, and nutrient availability (*Attayde and Hansson, 2001*; *Vanni, 2002*; *Vanni et al., 2006*). In this study, coculturing with aquatic animals (i.e. carp, crabs, and turtles) increased rice yield and N-use efficiency, and helped maintain soil N compared to the corresponding rice monoculture.

The aquatic animals had two important roles in these cocultured paddy ecosystems. One role involved competition, that is, aquatic animals reduced competitors (i.e. weeds) of rice plants and thereby enhanced rice yield. Some freshwater animals (e.g. carp, crabs, and crayfish) are omnivores that may consume some living organisms (e.g. weeds, algae, and phytoplankton) that compete with rice plants for nutrients. Carp and crabs, for example, can greatly reduce weeds in rice-carp systems (*Frei and Becker, 2005b*; *Xie et al., 2011*) and in rice-crab systems (*Lv et al., 2011*). Our field experiments also showed a reduction (45.3–51.9%) of weeds in the plots with carp, crabs, or turtles compared to the monoculture plots without herbicide use (*Figure 2*). Our dietary reconstruction based on stable-isotope data of $\delta^{13}C$ and $\delta^{15}N$ showed that carp and crabs obtained 34.8–50.2% of their total food from the rice field, including weeds, macro-algae, and phytoplankton (*Figure 3*); these results provide indirect evidence that carp and crabs reduced competitors of rice. Although some freshwater animals (e.g. turtles in our study) do not prefer to feed on weeds and other vegetative food sources, their activities disturb the paddy soil and thereby inhibit weed germination and growth (*Hu et al., 2016*; *He, 2017*).

A second role of aquatic animals concerned N, that is, aquatic animals increased the recycling of N in these cocultured paddy ecosystems. Many studies have shown that grazers can accelerate nutrient cycling in natural grassland and freshwater ecosystems by increasing nutrient availability in soil and nutrient-use efficiency of plants (*McNaughton et al., 1997*; *Atkinson et al., 2017*). In our study, the carp, crabs, and turtles obtained 50.2, 34.8, and 16.0%, respectively, of their food from the field rather than from the feed although sufficient feed was applied in our experiment to support the aquatic animals (*Figure 3*). Similar to the effects of grazers in the natural ecosystems, the aquatic animals (carp, crabs, and turtles) in the paddy ecosystem foraged on weeds, algae, phytoplankton, zooplankton, and benthic invertebrates that used N directly from the paddy field. Once these food source organisms are ingested, the aquatic animals convert them into biomass, feces, and excretions. Because ammonia represents from 75% to 85% of the N in aquatic animal excretions (*Chakraborty and Chakraborty, 1998*), the excretions can be directly utilized by rice plants. The aquatic animal feces may release nutrients once they are decomposed, or they may be stored in the form of soil organic matter. Thus, the promotion of N cycling by aquatic animals apparently explains, at least in part, why N-use efficiency of rice was higher and soil N content was more stable in the rice-aquatic animal coculture plots than in the monoculture plots (*Figure 1*).

In addition to reducing competition and increasing nutrient availability for rice plants via grazing, aquatic animals apparently increased nutrient availability for rice plants by enhancing organic matter decomposition. The percentage of maize leaves (added to the plots in 'litter tubes') that remained after 80 days (as indicated by the percentage of $^{13}C$ remaining) was significantly lower in the three RA treatments than in the RM treatment (*Figure 4*), indicating that carp, crabs, and turtles promoted organic matter decomposition in the field. Nutrients (e.g. N and P) in the organic matter (e.g. unconsumed feed, aquatic animal feces, and leaf litter) may be released by decomposition and then used by rice plants or other organisms in the field. Our tracing of feed-$^{15}N$ demonstrated that 13.0–35.1% of the feed-N was found in the rice plants (*Figure 5a*). These results suggested that the N in unconsumed or unassimilated feed was released via decomposition and was then used by the rice plants.

Unlike traditional rice-fish coculture systems in which no fish feed is applied (*Xie et al., 2011*), the current coculture systems, like those described in this study, often include the application of feed in order to obtain high aquatic animal yields (*Hu et al., 2016*). Whether such feed affects rice yield and soil fertility was also assessed in our study. The mesocosm experiments showed that 1.27 ± 0.09 g m$^{-2}$ (rice-carp), 0.36 ± 0.09 g m$^{-2}$ (rice-crab), and 1.12 ± 0.10 g.m$^{-2}$ (rice-turtle) of feed-$^{15}N$ accumulated

in the rice plant biomass (grain and straw) (*Figure 5b*). The mesocosm experiments also showed that feed-$^{15}$N accumulated in the soil (*Figure 6*). These results suggest that N use by cocultured rice and animals can be complementary and can increase N-use efficiency. The results also suggest that the unconsumed or unassimilated feed can function as a fertilizer for rice and can thereby increase the rice yield and the N content in the soil. In addition to N, phosphorus (P) also entered rice-animal coculture systems *via* feed in our study (*Appendix 2—table 1*). Like N, P is important for rice growth and yield (*Ahmed et al., 2017*). In our 4-year field experiments, the level of soil total P was similar at the end vs. the beginning under both rice monocultures and rice-animal cocultures (*Appendix 1—figure 3*), but rice yields were higher and more P was removed with the harvested products with coculture than with monoculture (*Appendix 2—table 2*). It follows that the P input via feed may contribute to the rice yield increase and the maintenance of soil P in the coculture systems.

The current results increase our understanding of how agricultural systems can use species diversity to increase sustainability. Planting diverse wild or crop species in field margins has been found to improve the management of crop pests and their natural enemies (*Bianchi et al., 2006*; *Tschumi et al., 2016*). Overyielding often occurs in intercropping systems when the coexisting crops benefit each other or when one benefits the others (*Snapp et al., 2010*; *Kremen et al., 2012*; *Li et al., 2014*; *Ren et al., 2014*; *Brooker et al., 2021*). In our study, the rice plants and the three aquatic animals (i.e. carp, crabs, and turtles) have a similar growing period and have similar water and temperature requirements, making it possible to develop a rice–aquatic animal partnership. Although carp, crabs, and turtles differ in biological traits and feeding activities, they play similar roles in increasing rice yield and N-use efficiency, and in stabilizing soil N.

Rice paddies provide food for half of the world's population (*Gross and Zhao, 2014*; *Edzesi et al., 2016*), and also provide other ecosystem services, including groundwater recharge, flood control, water purification, and the conservation of biodiversity, landscapes, and human cultures (*Bouman et al., 2007*; *Natuhara, 2013*; *Wang et al., 2018*). Modern rice farming currently faces the great challenge of how to increase yield while minimizing negative environmental effects (*Mueller et al., 2012*; *Chen et al., 2014*). Our study suggests that this challenge can be at least partially met by adding certain species of aquatic animals to rice monocultures. The resulting cocultures could produce more food (rice grain and fish) with less fertilizer and pesticides than rice monocultures. In our field experiment, an average annual aquatic animal yield ranging from 0.52 to 2.57 t ha$^{-1}$ was produced from the rice fields (*Figure 1*), suggesting that local farmers can obtain more income from their paddy fields. Moreover, the prices for grain and aquatic animal products from these cocultures were higher than from the local rice monoculture (*Hu et al., 2016*). Although costs of the cocultures are higher than the costs of monoculture because of the feed input and increased labor required for the management of two species, net income was still higher for cocultures than for monocultures because of the higher prices of the products and the reduced use of fertilizers and pesticides (*Hu et al., 2016*; *Wang et al., 2018*). Over the last two decades in China, the increased income has greatly increased farmer enthusiasm for the rice–aquatic animal cocultures (*Tang et al., 2020*).

While our current study and other previous studies have shown the positive effects of rice-aquatic animal coculture on rice production, farmer income, and soil N, possible negative effects resulting from the input of feed and the increased decomposition rate should be considered. These potential negative effects include eutrophication and increased carbon emission. Previous studies, however, found that rice-aquatic animal coculture would not cause serious N eutrophication in the field when the target aquatic animal yields were set below the following thresholds: 2.11 ± 0.22 t ha$^{-1}$ for rice-carp, 0.66 ± 0.08 t ha$^{-1}$ for rice-crab, and 3.62 ± 0.25 t ha$^{-1}$ for rice-turtle coculture systems (*Hu et al., 2013*; *Zhang et al., 2016*; *Hu et al., 2016*). The effects of rice-aquatic animal coculture on carbon emission (e.g. $CH_4$ emission) varied among reports (*Ding et al., 2020*; *Sun et al., 2021*). Some experiments indicated that $CH_4$ emissions were lower in rice–aquatic animal coculture systems than in rice monocultures (*Yuan et al., 2009*; *Sun et al., 2021*), while other experiments indicated that $CH_4$ emissions were higher in rice–aquatic animal coculture systems than in rice monocultures (*Frei and Becker, 2005a*; *Wang et al., 2019*). These differences in $CH_4$ emission could be caused by differences in aquatic animals, natural environments, and field management (*Ding et al., 2020*; *Dai et al., 2022*). In our study, the aquatic animals increased organic matter decomposition in the paddy field (*Figure 4*), suggesting that the release of nutrients but also of $CO_2$ would be higher with coculture than with monoculture.

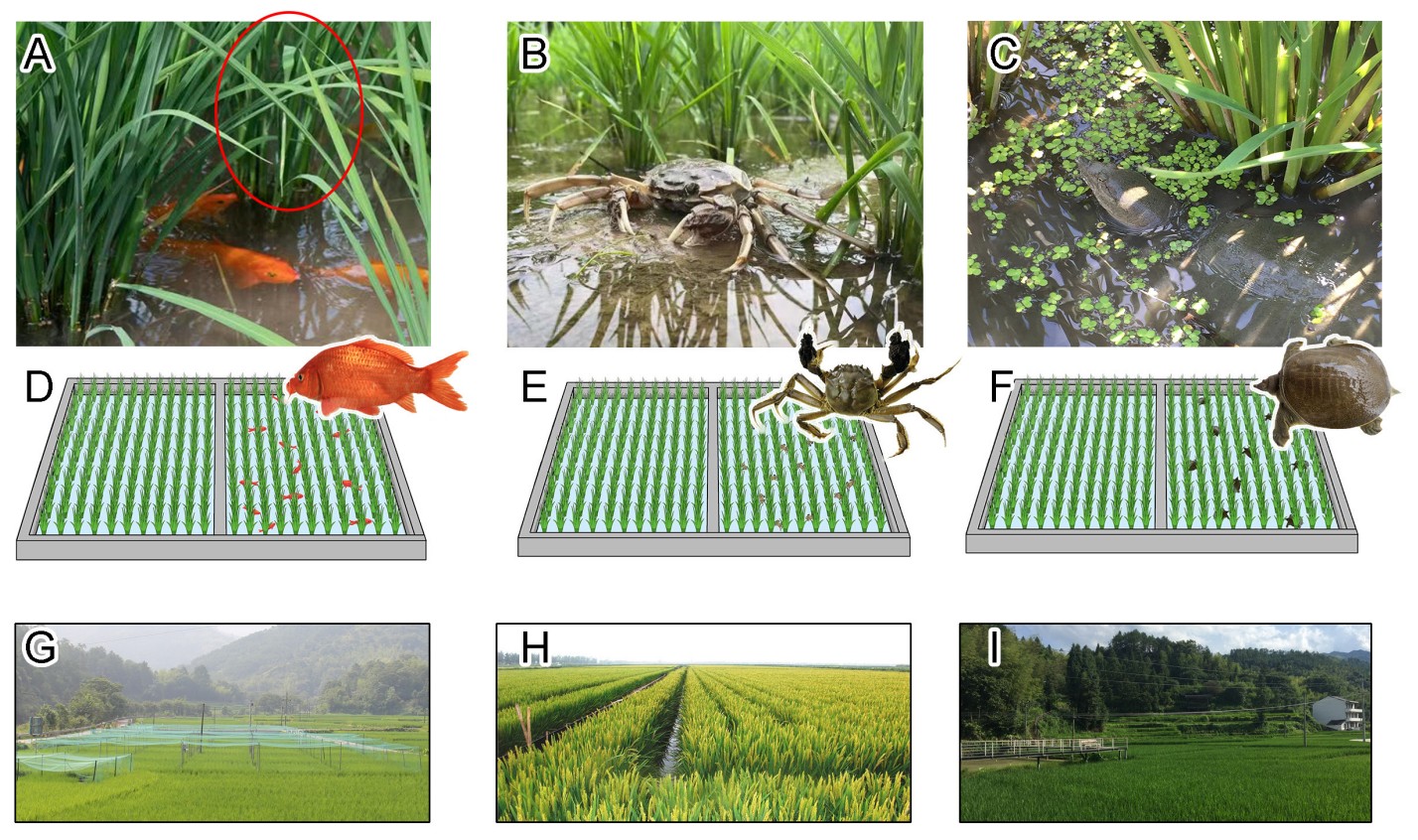

**Figure 7.** Illustration of the field experiments. The rice plants and carp (**A**), crab (**B**), and turtle (**C**) were photographed in the corresponding experimental plots (**D**, **E**, and **F**), which were arranged in completely randomized blocks in three rice-planting areas in Qingtian County (**G**), Panshan County (**H**), and Deqing County (**I**), respectively. The red circle in A indicates a rice hill. Photos of A, B and C were taken by Lufeng Zhao, Zhiming Li and Genlian Wang, respectively.

In our previous survey, the three typical types of coculture (i.e. rice-carp, rice-crab, and rice-turtle) were compared in terms of financial returns for the farmers. Financial returns were higher for rice-turtle systems (which is the most common type of coculture in south China) than for rice-carp and rice-crab systems (*Hu et al., 2016*; *Wang et al., 2018*). In addition to these three types of coculture, other types (e.g. rice-prawn/crayfish, rice-frogs, and rice-ducks) are also practiced in China and in other Asian countries (*Hu et al., 2015*; *Ahmed and Turchini, 2021*). In some areas, farmers apply feed to obtain higher animal yield from the cocultures, and the main components of the feed are residues of soybean after the oil has been extracted. Given the millions of hectares of rice fields with suitable conditions for rice-aquatic animal coculture, the cropland available for soybean production and the integration of rice-aquatic animal coculture with soybean production should be taken into account before substantial increases in coculture area are promoted.

## Materials and methods
### The rice–aquatic animal coculture systems in the study

Three coculture systems, namely rice-carp, rice-crab, and rice-turtle (*Figure 7*), were studied. These coculture systems have been developed and adapted to different rice-growing areas in China (*Hu et al., 2016*). The rice-carp coculture system, for example, has a long history and is widely practiced in south China (*Xie et al., 2011*; *Wang et al., 2018*). In recent decades, the rice-crab coculture system has been rapidly developing in northeastern China (*Xu et al., 2019*), and the rice-turtle coculture system has been rapidly developing in south China (*Zhang et al., 2017*). In these systems, the aquatic animals (i.e. carp, crabs, and softshell turtles, *Figure 7*) are partnered with rice plants during the whole rice-growing period (130–150 days) and are harvested every year when the rice matures. These three

aquatic animals are economically important, and are usually cultured in fish ponds or paddy fields by local farmers. The meat of all these species is a popular food of the local people. To increase the growth and quality of aquatic animals in coculture systems, farmers often apply feed in the form of pellets (*Appendix 2—table 1*).

The common carp (*Cyprinus carpio*) has adapted to paddy habitats and is predominantly used for rice-carp coculture (*Figure 7A*, *Xie et al., 2011*; *Wang et al., 2018*). The growing season for rice-carp systems is usually from late May to early October (ca. 125 days). Carp fry (ca. 40 g each) are often released into the rice field 1 week after rice is transplanted. The carp, which do not feed on rice plants, live in the paddy field until the rice plants are mature, at which time the carp are harvested. In the current study, pellet feed (5.37% N) was applied each day at ca. 6:30 am. The daily amount of feed was initially set as 4% of the fish fresh weight and was increased by 3% every 10 days afterwards. By carp harvest, the total inputs of the feed and feed-N for RA plots were 1.46 t ha$^{-1}$ and 78.41 kg ha$^{-1}$, respectively (*Appendix 2—table 1*). After they are harvested from paddy fields, the carp are directly used as food or are temporary cultured in fish ponds until they are used as food.

The Chinese mitten crab (*Eriocheir sinensis* Milne-Edwards) is used in the rice-crab system and is also well adapted to the paddy environment (*Figure 7B*, *Xu et al., 2019*). In the rice-crab system, the growing season is from middle May to middle October (ca. 150 days). The juvenile crabs (ca. 1.15 g each) are released into rice fields 1 week after rice is transplanted. The juvenile crabs do not feed on rice plants and 'live' together with rice plants until harvest. The crabs molt several times during their time in the paddy field. In the current study, pellet feed (4.52% N) was applied once each day at ca. 6:30 am. The daily amount of feed was initially set as 3–5% of the crab fresh weight and was increased by 3% every 10 days afterwards. By crab harvest, the total inputs of the feed and feed-N for RA plots were 1.78 t ha$^{-1}$ and 80.46 kg ha$^{-1}$, respectively (*Appendix 2—table 1*). After they are harvested from the paddy fields, the crabs are used as food or are placed in ponds and are used as a source of crabs for the next year.

The Chinese softshell turtle (*Pelodiscus sinensis*) is often cultured in paddy fields by local farmers in southeastern China (*Figure 7C*). The turtles are omnivores but prefer animal diets (*He, 2017*). In this rice-turtle system, the growing season is from middle June to early November (ca. 140 days). The baby turtles (ca. 150 g each) are released into the field 1 week after rice is transplanted. The turtles remain with the rice plants for the whole rice-growth period. In the current study, pellet feed (6.51% N) was applied twice per day at ca. 7:00 am and 5:00 pm throughout the coculture period. The daily amount of feed was initially set as 0.5%–1.0% of the turtle fresh weight but was increased as the turtles grew. By turtle harvest, the total inputs of the feed and feed-N for RA plots were 1.62 t ha$^{-1}$ and 105.46 kg ha$^{-1}$, respectively. (*Appendix 2—table 1*). After they are harvested from paddy fields, the turtles are used as food or are temporarily cultured in fish ponds until they are used as food.

## Ethics statement

In the following experiments, the samples of all aquatic animals (i.e. carp, crabs, and softshell turtles) were collected and measured in accordance with the approved guidelines of the Zhejiang University Experimental Animal Management Committee (reference number SYXK(Zhe)2018–0016). Details on the handling of animal samples were described in the Methods section.

## Field experiments

Each of three field experiments (one each for rice-carp, rice-crab, and rice-turtle systems) was conducted for 4 years (2017–2020) at a site (one system per site) where the particular system was widely practiced. The three sites are described in the supporting information (see Appendix 3).

All three experiments used a completely randomized block design with two treatments: rice monoculture (RM) and rice-aquatic animal coculture (RA) with feed addition for aquatic animal (*Appendix 2—table 1*). Each treatment at each site was represented by six replicate plots with a size of 0.01 ha per plot for rice-carp and rice-crabs, and 80 m$^2$ per plot for rice-turtles. No chemicals were used to control weeds, pests, or diseases during the experiments. The plot size and detailed experimental procedures are described in Appendix 4.

In 2018, the stable-isotope $^{13}$C labeling method was used to determine the organic matter decomposition rate in the three field experiments (*Cheng et al., 2012*). 'Litter tubes' containing maize (*Zea

*mays* L.) leaves enriched with $^{13}C$ ($\delta^{13}C$ –13.6‰) were used as described in Appendix 5. The percentage of $^{13}C$ remaining in the litter tubes after 40 and 80 days represented the decomposition rate.

We used a stable isotope ($^{13}C$ and $^{15}N$) technique to determine how the aquatic animals (i.e. carp, crabs, or turtles) used food resources (e.g. weeds, zoobenthos, zooplankton, and phytoplankton) in the field (*Michener and Schell, 1994*). During the rice growing period in 2019, we collected living organisms that were consumed by the aquatic animals in each of the three experiments (*Caut et al., 2009*; *Guo et al., 2016*) (see Appendix 6). Before and after the experiments, we also collected muscles from the aquatic animals (carp, crabs, or turtles) (see Appendix 7).

We ground all dried samples of food sources and aquatic animals and analyzed their isotopic δ value and content ($^{13}C$ and $^{15}N$). The δ value was calculated as ($[R_{sample}/R_{standard}]$−1) × 1000, where R represents $^{13}C$:$^{12}C$ or $^{15}N$:$^{14}N$ (*Peterson and Fry, 1987*). Dietary contributions of input feed and potential food sources from the rice field were analyzed by stable isotopic dietary reconstruction with the R package 'siar' (*Phillips and Gregg, 2003*; *Phillips et al., 2005*; *Parnell and Jackson, 2011*; *R Core Team, 2021*). The discrimination factors of $^{13}C$ and $^{15}N$ for carp, crabs, and turtles in dietary reconstruction were previously determined (see Appendix 8).

Three weeks before rice was harvested in each experimental year (from 2017 to 2020), 5 quadrats (1 m²) were randomly placed in each plot to evaluate weed infestation. For each quadrat, the aboveground dry weed biomass was measured.

In each experimental year (from 2017 to 2020), rice and aquatic animals were harvested from the whole experimental plots when rice plants matured. Rice yield was determined by manually harvesting entire plots. Rice grain was air-dried and weighed. Rice yield was estimated as tons of air-dried grain per ha. One week before rice was harvested, aquatic animals were collected from entire plots by driving them into the corner of field as the water was drained. Yield was expressed as tons of fresh aquatic animal biomass per ha.

At harvest in each year of the experiments, samples of rice plants and aquatic animals were collected for determining N content. Five hills of rice were collected in each plot. The separated grain and straw were oven-dried at <65°C to a constant weight. The aquatic animal samples were kept in water for 24 hr to permit the emptying of intestinal contents. The clean aquatic animal samples were oven-dried at 105°C to a constant weight. Both rice and aquatic animal samples were ground with a ball mill (RETSCHMM 400, Germany). The N content in rice straw and grain and in the aquatic animals was measured by the Kjeldahl method (*Bremmer and Mulvaney, 1982*).

Every experimental year, soil samples (0–20 cm depth) were collected immediately after harvest from each plot. All soil samples were air dried. Soil organic matter (SOM) content was determined by the $K_2Cr_2O_7$ oxidation method, and total nitrogen (N) content was determined by the Kjeldahl method (*Lu, 1999*).

We used the data collected in 2018 to estimate apparent N-use efficiency (ANUE) by calculating percentage of the input N that was used by rice and aquatic animals (*Moll et al., 1982*; *Mayer et al., 2015*; *Zhang et al., 2017*) as follows:

$$\text{ANUE } (\%) = \frac{N_y}{N_s} \times 100 \qquad (1)$$

where $N_y$ is the total amount of N contained in the grain and straw of rice plants, and in the aquatic animals that were removed from the paddy system, and $N_s$ is the total input of fertilizer-N and feed-N. $N_y$ was determined by multiplying the biomass of rice (grain and straw) and aquatic animals by the percentage of N in rice and aquatic animals. We assumed that the natural N input (e.g. N fixed by bacteria, N in the irrigation water, and atmospheric N deposition) was similar between RM and RA plots, and natural N input was therefore not included in our estimations of ANUE.

Statistical analysis was conducted using the GLM in SPSS (V.20.0, RRID: SCR_002865). All data were subjected to a homogeneity test. If the data did not meet the assumptions of normality and homogeneity, they were log-transformed before analysis. For each field experiment, ANOVAs with a split-plot design (i.e. treatment RM and RA as the main plots and experimental years as the sub-plots) were performed on rice yields, total soil N content, ANUE, and weed biomass. For RM or RA plots, total N in the soil at the beginning and end of the experiment were compared by using paired *t*-tests (SPSS V.20.0, RRID: SCR_002865).

## Mesocosm experiments

To determine whether unconsumed and unassimilated feed-N is used by aquatic animals in rice-carp, rice-crab, or rice-turtle systems, we conducted three independent mesocosm experiments (one for each kind of system) at the Experimental Station of Zhejiang University in Deqing County, Zhejiang Province (30°33′N, 119°32′E.). The mesocosm experiments were conducted in 2019. The fate of feed-N was traced by using stable-isotope $^{15}$N-labeled feed in each mesocosm experiment.

For rice-carp and rice-crab experiments, each mesocosm was a cylindrical, 1017 L plastic stock tank. Intact soil (300 L) from a corresponding rice field was added to each mesocosm to a depth of 30 cm. The mesocosms were placed in a rice paddy field (the corresponding fields that were used for the three field experiments) so that their bottoms were 20 cm below the soil surface, and the top of each microcosm was above the water line. For the rice-turtle system, 2 m × 2 m plots were established in a rice paddy field. Each plot had an independent water inlet and outlet and was separated from water mixing by concrete ridges.

Each of the three mesocosm experiments had a completely randomized block design with six replicate blocks. The treatments were rice monoculture (RM) and rice-aquatic animals coculture (RA). For the RM and RA, varieties of rice and aquatic animal species were the same as in the field experiment. The detailed procedures for each mesocosm experiment are provided in Appendix 9.

Because soybean is the major raw ingredient of feeds for the three aquatic animals (carp, crabs, and turtles), we first labeled soybean powder with $^{15}$N (see Appendix 10). We then mixed the labeled soybean powder with the general feed of carp, crabs, and turtles as indicated in *Appendix 2—table 3*.

At harvest, samples of rice plants, aquatic animals (carp, crabs, and turtles), and soil were collected from each mesocosm (see Appendix 11). The $^{15}$N content in all samples of rice plants, aquatic animals, and soil was quantified with a ThermoFinnigan DELTA Plus continuous flow isotope ratio mass spectrometer.

We calculated the contribution of feed-N to total rice biomass-N with a linear mixing model (*Phillips and Gregg, 2003*):

$$\delta^{15}N_{RM} \times a + \delta^{15}N_{feed} \times b = \delta^{15}N_{RA} \tag{2}$$

$$a + b = 1 \tag{3}$$

$$b = (\delta^{15}N_{RA} - \delta^{15}N_{RM})/(\delta^{15}N_{feed} - \delta^{15}N_{RM}) \tag{4}$$

where a is the contribution of soil N to rice total biomass-N; b is the contribution of feed-N to rice total biomass-N; $\delta^{15}N_{RM}$ is the $\delta^{15}$N value of the rice plants in the RM treatment; $\delta^{15}N_{feed}$ is the $\delta^{15}$N value of $^{15}$N-labeled feed; and $\delta^{15}N_{RAS}$ is the $\delta^{15}$N value of rice plants in the RA treatment. For RM or RA mesocosms, $\delta^{15}$N and total N in the soil at the beginning and end of the experiment were compared by using one-tailed *t*-tests under the assumption that $\delta^{15}$N and total N in the soil would increase after the experiments (SPSS V.20.0, RRID: SCR_002865).

## Acknowledgements

We thank Minfang Wu, Yongqing Yu, and Genlian Wang for their assistance with the field experiments.

## Additional information

### Funding

| Funder | Grant reference number | Author |
| --- | --- | --- |
| Natural Science Foundation of China | U21A20184 | Xin Chen |
| Science and Technology Program of Zhejiang Province | 2022C02058 | Xin Chen |

| Funder | Grant reference number | Author |
|---|---|---|
| Science and Technology Program of Zhejiang Province | LGN22C030002 | Liangliang Hu |
| Natural Science Foundation of China | 31661143001 | Xin Chen |
| Natural Science Foundation of China | 31770481 | Xin Chen |

The funders had no role in study design, data collection and interpretation, or the decision to submit the work for publication.

## Author contributions

Liang Guo, Formal analysis, Investigation, Methodology, Software, Writing – original draft; Lufeng Zhao, Junlong Ye, Zijun Ji, Keyu Bai, Sijun Zheng, Liangliang Hu, Formal analysis, Investigation, Methodology, Software; Jian-Jun Tang, Formal analysis, Investigation, Methodology, Resources; Xin Chen, Conceptualization, Methodology, Supervision, Validation, Writing – original draft, Writing - review and editing

## Author ORCIDs

Keyu Bai ⓘ http://orcid.org/0000-0003-3644-0844
Xin Chen ⓘ http://orcid.org/0000-0001-6622-1074

## Ethics

In the following experiments, the samples of all aquatic animals (i.e., carp, crabs, and softshell turtles) were collected and measured in accordance with the approved guidelines of the Zhejiang University Experimental Animal Management Committee. Details on the handling of animal samples were described in the Methods section.

## Decision letter and Author response

Decision letter https://doi.org/10.7554/eLife.73869.sa1
Author response https://doi.org/10.7554/eLife.73869.sa2

---

# Additional files

## Supplementary files

- Appendix 1—figure 1—source data 1. Related to data in *Appendix 1—figure 1*.
- Appendix 1—figure 2—source data 1. Related to data in *Appendix 1—figure 2*.
- Appendix 1—figure 3—source data 1. Related to data in *Appendix 1—figure 3*.
- Transparent reporting form

## Data availability

All data generated or analyzed during this study are included in the manuscript and supporting files. Source data files have been provided for Figures 1–6 and Appendix 1—figure 1–3.

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

## Appendix 1

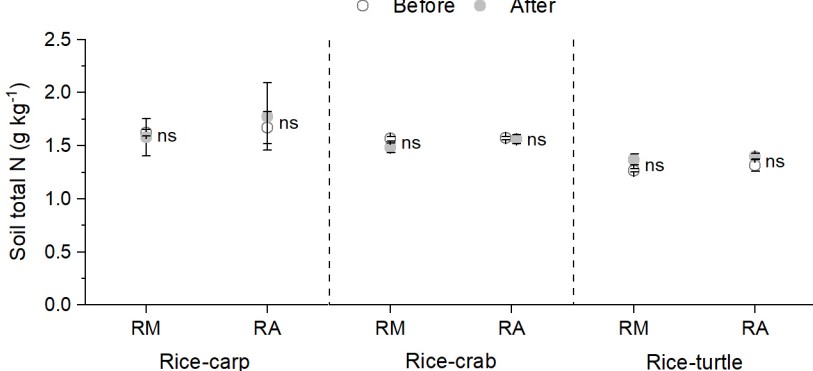

**Appendix 1—figure 1.** Total soil N content in rice monoculture (RM) and coculture treatments (RA) before and after the field experiments. RM: rice monoculture; RA: rice coculture with an aquatic animal. Values are means ± SE (n = 6), ns indicates no significant difference between RM and RA (p > 0.05).

The online version of this article includes the following source data for appendix 1—figure 1:

- **Appendix 1—figure 1—source data 1.** Related to data in *Appendix 1—figure 1*.

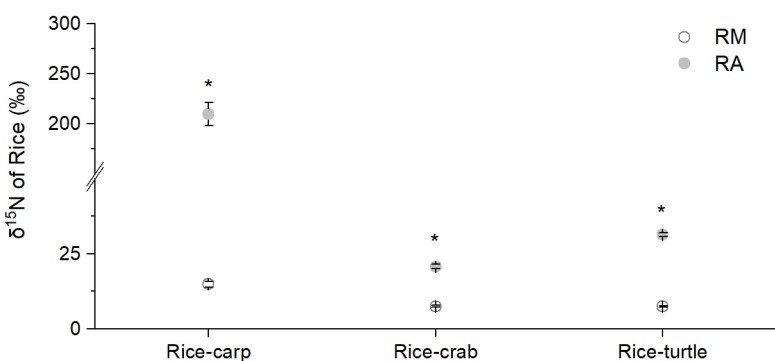

**Appendix 1—figure 2.** $\delta^{15}N$ of rice shoots in rice monoculture (RM) and coculture treatments (RA) in the mesocosm experiments. RM: rice monoculture; RA: rice coculture with an aquatic animal. Values are means ± SE (n = 6), * indicates a significant difference between RM and RA (p < 0.05).

The online version of this article includes the following source data for appendix 1—figure 2:

- **Appendix 1—figure 2—source data 1.** Related to data in *Appendix 1—figure 2*.

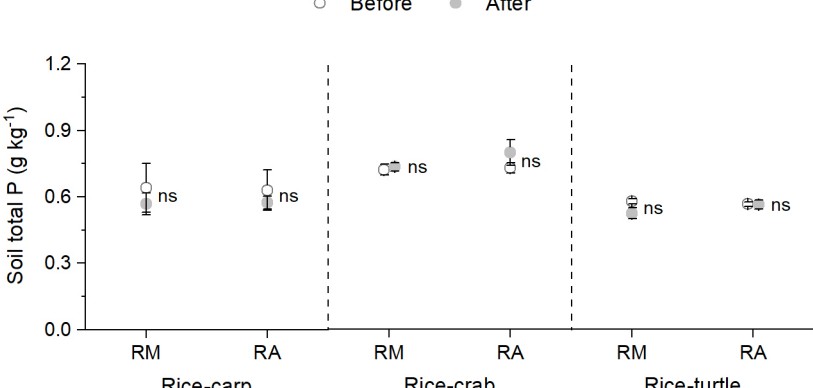

**Appendix 1—figure 3.** Total soil P content in rice monoculture (RM) and coculture treatments (RA) before and after the field experiments. RM: rice monoculture; RA: rice coculture with an aquatic animal. The P contents in soil were analyzed with a San$^{++}$ Continuous Flow Analyzer (Skalar, Netherlands) after the air-dried soil samples were and digested by the $K_2SO_4$-$CuSO_4$-Se method (**Lu, 1999**). For RM or RA plots, total P in the soil at the beginning and end of the experiment were compared by using paired t-tests (SPSS V.20.0, RRID: SCR_002865). Values are means ± SE (n = 6), ns indicates no significant difference between RM and RA (p > 0.05).

The online version of this article includes the following source data for appendix 1—figure 3:

• **Appendix 1—figure 3—source data 1.** Related to data in *Appendix 1—figure 3*.

# Appendix 2

**Appendix 2—table 1.** Input of N and P into rice-animal coculture systems via feed and fertilizer application.

| System | Treatment | Feed | Fertilizer* | | | Feed | Fertilizer | | |
|--------|-----------|------|-------------|--|--|------|------------|--|--|
| | | | Base fertilizer | Top-dress fertilizer | Total | | Base fertilizer | Top-dress fertilizer | Total |
| | | | **Nitrogen input (kg ha⁻¹)** | | | | **Phosphorus input (kg ha⁻¹)** | | |
| Rice-carp | RM | --- | 82.50 | 29.40 | 111.90 | --- | 36.02 | 5.31 | 41.33 |
| | RA | 78.41 | 82.50 | --- | 82.50 | 19.63 | 36.02 | --- | 36.02 |
| Rice-crab | RM | --- | 112.5 | 44.94 | 157.44 | --- | 49.12 | 11.79 | 60.91 |
| | RA | 80.46 | 112.5 | --- | 112.50 | 20.62 | 49.12 | --- | 49.12 |
| Rice-turtle | RM | --- | 82.5 | 44.94 | 127.44 | --- | 36.02 | 11.79 | 47.81 |
| | RA | 105.46 | 82.5 | --- | 82.50 | 36.61 | 36.02 | --- | 36.02 |

*The total amount of feed-N and - P input were determined by multiplying the amount of feed by the percentage of N and P in feed that were analyzed with a San⁺⁺ Continuous Flow Analyzer (Skalar, Netherlands) (**Lu, 1999**).

**Appendix 2—table 2.** The total P removed with the harvested products under rice monoculture and coculture in the field experiment.
Values are means ± SE (n = 6)

| System | Treatment | Rice grain* (kg ha⁻¹) | Rice straw* (kg ha⁻¹) | Aquatic animal* (kg ha⁻¹) | Total (kg ha⁻¹) |
|--------|-----------|------------------------|------------------------|----------------------------|------------------|
| Rice-carp | RM | 18.53 ± 0.92 | 5.49 ± 0.25 | --- | 24.02 ± 1.15 |
| | RA | 21.68 ± 0.50 | 6.44 ± 0.32 | 7.45 ± 0.95 | 35.57 ± 0.96 |
| Rice-crab | RM | 26.38 ± 1.53 | 8.60 ± 0.67 | --- | 34.98 ± 1.87 |
| | RA | 34.74 ± 2.19 | 13.67 ± 1.04 | 1.07 ± 0.15 | 49.48 ± 2.79 |
| Rice-turtle | RM | 19.57 ± 0.72 | 12.64 ± 0.44 | --- | 32.21 ± 0.42 |
| | RA | 24.19 ± 1.17 | 17.19 ± 0.69 | 20.66 ± 1.47 | 62.04 ± 2.60 |

*The total amount of P contained in the grain and straw of rice plants, and in the aquatic animals were determined by multiplying the biomass of rice (grain and straw) and aquatic animals by the percentage of P in rice and aquatic animals that were analyzed with a San⁺⁺ Continuous Flow Analyzer (Skalar, Netherlands) (**Lu, 1999**).

**Appendix 2—table 3.** Properties of original feed and ¹⁵N-labeled feed for the three field experiments.
The original feed (in which most N was supplied by soybean powder) was supplemented with ¹⁵N-labeled soybean powder to obtain ¹⁵N-labeled feed.

| Experiment | Original feed | | ¹⁵N-labeled soybean powder | | ¹⁵N-labeled feed | | % of labeled soybean powder in the ¹⁵N-labeled feed |
|------------|---------------|----------|-----------------------------|----------|-------------------|----------|-----------------------------------------------------|
| | N% | δ¹⁵N | N% | δ¹⁵N | N% | δ¹⁵N | |
| Rice-carp | 5.58 | 1.73 | 6.61 | 1,539 | 5.25 | 649 | 33 |
| Rice-turtle | 8.21 | 8.41 | 6.98 | 445 | 7.54 | 128 | 30 |
| Rice-crab | 5.94 | 1.19 | 6.98 | 445 | 7.27 | 145 | 33 |

## Appendix 3

### Sites of the field experiments

The rice-carp (common carp, *Cypinius carpia*) field experiment was conducted in Qingtian County (120°16'02"E, 28°01'24"N), Southeast Zhejiang Province. The rice-carp system in that hilly and mountainous county has a history of more than 1000 years and exclusively cultures an indigenous breed of common carp, which was used for our experiment. The experimental site has a subtropical monsoon climate (mean annual temperature: 17.5 °C, mean annual rainfall: 1432 mm) and an acidic sandy loam soil (total organic matter: 30.72–32.92 g kg$^{-1}$, total N: 2.09–2.79 g kg$^{-1}$, pH: 5.4).

The rice-crab (Chinese mitten crabs, *Eriocheir sinensis*) field experiment was conducted in Panshan County, Liaoning Province (41°9'24"N, 122°15'1"E). Because of its proximity to the Bohai Sea, this province provides natural habitats for the breeding and growing of Chinese mitten crabs. The experimental site has a monsoon continental climate (mean annual rainfall: 620 mm, mean annual temperature: 9.2°C) and a loamy soil (total organic matter: 23.08–29.53 g kg$^{-1}$, total N: 1.42–1.66 g kg$^{-1}$, pH: 8.2).

The rice-turtle (soft-shelled turtle, *Pelodiscus sinensis*) field experiment was conducted in Deqing County, Northern Zhejiang Province (120°7'30"E, 30°37'42"N). The soft-shelled turtle used for the experiment is indigenous breed to that flat, open county. The experimental site has a subtropical monsoon climate (mean annual rainfall: 1379 mm, mean annual temperature: 14°C) and a sandy loam soil (total organic matter: 30.72–32.92 g kg$^{-1}$, total N: 2.09–2.79 g kg$^{-1}$, pH: 5.4).

## Appendix 4

### Field experiment details

For the rice-carp field experiment, we randomly assigned one RM plot and one RA plot (0.01 ha per plot) to each of six blocks. We also separated field plots by 50-cm-high concrete brick walls. Independent water inlets and outlets were setup for each plot to prevent interference between plots. Rice (ZhongZheYou No. 8) was transplanted 4 weeks after germination, with a spacing of 30 cm × 30 cm (one seedling per hill). Two days before rice transplanting, a compound fertilizer (15% N, 15% $P_2O_5$, 15% $k_2O$) used as basal fertilizer was broadcast at the rate of 550 kg ha$^{-1}$ for both treatments. No top-dress fertilizer was applied for RA, but urea (46% N) at the rate of 37.5 kg ha$^{-1}$ and compound fertilizer (15% N, 15% $P_2O_5$, 15% $K_2O$) at the rate of 81 kg ha$^{-1}$ were used as top-dress fertilizer for RM during the rice growth period (*Appendix 2—table 1*). No pesticide was applied for both treatments. During the rice growth period, all plots were continuously flooded at about 20 cm depth. Six days after rice transplanting, we released 60 carp fry (ca. 40 g each, purchased from a local farmer) into each RA plot. Pellet feed (5.37% N) was added each day at ca. 6:30 am to feed the carp fry. The daily amount of feed was initially set as 4% of the fish fresh weight and was increased by 3% every 10 days afterwards. By carp harvest, the total inputs of the feed and feed-N for RA plots were 1.46 t ha$^{-1}$ and 78.41 kg ha$^{-1}$, respectively.

For the rice-crab experiment, we randomly assigned one RM plot and one RA plot (0.01 ha per plot) to each of six blocks. We also separated the plots with 50-cm-high PVC boards. Independent water inlets and outlets were setup for each plot to prevent interference between plots. Rice plants (Yanfeng No. 47) were transplanted into each plot 4 weeks after germination at a spacing of 30 cm × 30 cm (one seedling per hill). Two days before rice transplanting, a compound fertilizer (15% N, 15% $P_2O_5$, 15% $k_2O$) used as basal fertilizer was broadcast at the rate of 750 kg·ha$^{-1}$ for both treatments. No top-dress fertilizer was applied for RA, but urea (46% N) at the rate of 39 kg ha$^{-1}$ and compound fertilizer (15% N, 15% $P_2O_5$, 15% $K_2O$) at the rate of 180 kg ha$^{-1}$ were used as top-dress fertilizer for RM during the rice growth period (*Appendix 2—table 1*). No pesticide was applied for both treatments. One week after rice transplanting, 375 g of juvenile crabs (ca. 1.15 g each) were released into each RA plot. During the experiment, pellet feed (4.52% N) was applied once each day at ca. 6:30 am. The daily amount of feed was initially set as 3%–5% of the crab fresh weight and was increased by 3% every 10 days afterwards. By crab harvest, the total inputs of the feed and feed-N for RA plots were 1.78 t ha$^{-1}$ and 80.46 kg·ha$^{-1}$, respectively.

For the rice-turtle experiment, we randomly assigned one RM plot and one RA plot (8 m × 10 m per plot) to each of six blocks. The plots were separated by 1.5-m-high concrete ridges. Independent water inlets and outlets were setup for each plot to prevent interference between plots. Rice (Qing-Xi No. 8) was transplanted into each plot 4 weeks after germination, at a spacing of 30 cm × 30 cm (one seedling per hill). Two days before rice transplanting, a compound fertilizer (15% N, 15% $P_2O_5$, 15% $k_2O$) used as basal fertilizer was broadcast at the rate of 550 kg ha$^{-1}$ for both treatments. No top-dress fertilizer was applied for RA, but urea (46% N) at the rate of 39 kg ha$^{-1}$ and compound fertilizer (15% N, 15% $P_2O_5$, 15% $K_2O$) at the rate of 180 kg ha$^{-1}$ were used as top-dress fertilizer for RM during the rice growth period (*Appendix 2—table 1*). No pesticide was applied for both treatments. One week after rice transplanting, 8 young turtles (ca. 150 g each) were released into each RA plot. Pellet feed (6.51% N) was applied twice per day at about 7:00 am and 5:00 pm throughout the experiment. The daily amount of feed was initially set as 0.5%–1.0% of the turtle fresh weight but was increased as the turtles grew. By turtle harvest, the total inputs of the feed and feed-N for RA plots were 1.62 t ha$^{-1}$ and 105.46 kg ha$^{-1}$, respectively.

# Appendix 5

## Method for testing organic matter decomposition rate

As a C4 plant, maize (*Zea mays*) is naturally [13]C-enriched as compared to C3 plants like rice. We therefore used maize tissue to determine the decomposition rate of organic matter in our experiment. Maize was planted in a loam soil and was harvested just before flowering. Maize leaves were dried and cut into pieces ( < 1.0 cm$^2$). About 0.5 g of the leaf segments, mixed with 50 g of sands, was added to a tube (10 cm long, 5 cm in diameter), the ends of which were covered with a 20 μm mesh; this resulted in a "litter tube", which was analogous to a litter bag. The initial δ[13]C contents in the litter tubes are listed in *Appendix 5—table 1*.

**Appendix 5—table 1.** Initial δ[13]C contents in materials used to test the decomposition rate in the three field experiments.

| Experiment | C (%) | δ[13]C | [13]C (mg) |
|---|---|---|---|
| rice-carp | 0.41 | −16.21 | 2.18 |
| rice-crab | 0.44 | −16.24 | 2.39 |
| rice-turtle | 0.43 | −15.62 | 2.33 |

   In each plot, we randomly buried 20 tubes at 0–10 cm soil depth in the plant row and harvested them after 40 and 80 days. On each sampling date, we retrieved five tubes from each plot. The material (sample) in the tubes was air-dried and ground (RETSCHMM 400, Germany). Carbon content and the ratio of [13]C to [12]C were determined using a continuous flow isotope ratio mass spectrometer (CF-IRMS, ThermoFinnigan DELTA Plus, Waltham, MA, USA). The [15]C remaining was expressed as a percentage of [15]C in the initial materials.

## Appendix 6

### Method for sample collection of aquatic animal food sources

In 2019, we collected all possible aquatic animal food sources in the field, including weeds, algae, azollaceae, phytoplankton, zooplankton, zoobenthos, and particulate organic matter (POM). Weeds, algae, and azollaceae were collected, rinsed, and oven-dried. Zoobenthos collected from 0 to 20 cm topsoil were placed in distilled water for 48 hr. Zooplankton were collected from field water with a 0.064 mm nylon net. To collect phytoplankton and POM, field water was first passed through the nylon net; the phytoplankton and POM were then collected on glass fiber filters (GF/C, Whatman, pre-combusted at 450°C) via vacuum-filtration in the laboratory. All samples were oven dried at 50°C and stored until analyzed.

## Appendix 7

### Method of aquatic animal sample collection

Before and after the experiments in 2018, carp, crabs, or turtles were randomly sampled and stored at –20°C. Muscles were peeled from every animal sample, oven dried at 50°C, and separated into two sub-samples. One sub-sample was directly used for $^{15}$N analysis, and the other was used for $^{13}$C analysis after being pretreated with 2:1 chloroform and methanol for lipid removal.

## Appendix 8

### Determination of the discrimination factors of $^{13}$C or $^{15}$N

In the dietary reconstruction, the discrimination factor ($\triangle$) is the difference in $\delta^{13}$C or $\delta^{15}$N values between food sources and food consumers (*Peterson and Fry, 1987*). In our study, $\triangle^{15}$N and $\triangle^{13}$C were 2.73‰ and 1.71‰, respectively, for carp (*Hu et al., 2015*), and were 2.75‰ and 0.75‰, respectively, for crabs and turtles (*Caut et al., 2009*).

## Appendix 9

### Mesocosm experiments

For the rice-carp mesocosm experiment, five hills of rice plants (one seedling per hill) were planted in each mesocosm. One week after rice transplanting, five fish fry (ca. 10 g each) were added to each mesocosm. A water level of 25 cm was maintained during the experiment, with pH 6.0–6.5, $NO_2^- < 0.01$ mg $L^{-1}$, and $NH_4^+ < 0.2$ mg $L^{-1}$. No fertilizer was applied. At 4 days after the fry were introduced and on every day of the experiment thereafter, the $^{15}N$-labeled feed was applied twice per day (at 7:00 and 17:00). Throughout the experiment, the total amount of feed applied was 126 g $m^{-2}$.

For the rice-crab mesocosm experiment, four hills of rice plants (three rice seedlings per hill) were transplanted into each mesocosm. One week after rice transplanting, 50 juvenile crabs (ca. 1.22 g each) were released into each mesocosm. A water level of 25 cm was maintained during the experiment, with pH 8.0–8.5, $NO_2^- < 0.1$ mg $L^{-1}$, and $HN4^+ < 0.5$ mg $L^{-1}$. At 3 days after crab release and on every day of the experiment thereafter, the $^{15}N$-labeled feed was applied once per day at 18:00. Throughout the experiment, the total amount of feed applied was 124 g $m^{-2}$.

For the rice-turtle mesocosm experiment, four hills of rice plants (three rice seedlings per hill) were transplanted into each mesocosm. One week after rice transplanting, 14 young turtles (ca. 3.5 g each) were released into each mesocosm. A water level of 30 cm was maintained during the experiment, with pH 8.0–8.5, $NO_2^- < 0.1$ mg $L^{-1}$, and $NH_4^+ < 0.5$ mg $L^{-1}$. Four days after turtle release and on every day of the experiment thereafter, the $^{15}N$-labeled feed was applied twice each day (at 6:00 and 16:00). Throughout the experiment, the total quantity of feed applied was 130 g $m^{-2}$.

## Appendix 10

## Method of obtaining soybean powder labeled with $^{15}$N

Soybean plants were planted in pots in a green house. On 3 days during the growing period, $^{15}$N-enriched $(NH_4)_2SO_4$ (99.7% $^{15}$N) was added to the soil at the rate of 5 mg N kg$^{-1}$ soil. After they were harvested, soybeans were oven-dried at 65°C and ground into powder, which was used as the $^{15}$N labeling ingredient in the $^{15}$N-enriched feed.

## Appendix 11

### Sample collecting and preparing for rice, aquatic animals, and soil

At harvest, the aboveground portions of rice plants on one hill from each mesocosm were separated into grain and straw and were oven-dried at 65 °C to a constant weight. Samples of carp, crabs, or turtles were randomly collected from each mesocosm. These aquatic animal samples were placed in clean water for 48 hr to empty their guts and were stored at –80 °C after being freeze-dried. Soil samples (0–20 cm depth) were collected from each mesocosm before and after the experiment and were air-dried.

