## [Editor Report]

This is a well conducted experimental study in which rice monocultures are compared with rice-aquatic animal co-cultures at multiple sites over multiple years. Co-culture increases plant yield and adds animal yield, but requires extra input of animal feed. Overall co-culture benefits yield and sustainability.

---

## [Decision Letter]

**Decision letter after peer review:**

Thank you for submitting your article "Using aquatic animals as partners to increase yield and soil nitrogen in the paddy ecosystem" for consideration by *eLife*. Your article has been reviewed by 4 peer reviewers, including Bernhard Schmid as the Reviewing Editor and Reviewer #1, and the evaluation has been overseen by Detlef Weigel as the Senior Editor.

Essential Revisions:

1) Statistical analysis: it seems this was done correctly, e.g using years as subplots and not as independent measurements. However, this should be explained in more detail by presenting nominator and denominator degrees of freedom for F-tests and number of replicates in figures or legends.

2) Report amount of N (and P?) input via animal feed and arriving in soil. Discuss this in relation to other N in system and potential consequences of added P.

3) Discuss broader consequences of rice-animal co-culture in terms of economic costs/benefits to farmer, potential side effects on land used to produce animal feed and on nutrient and carbon cycle (including greenhouse-gas emissions).

4) Explain in more detail how animals were treated, grew (one or several years), affected rice plants and were "harvested" (every year all animals?).

5) It would be nice to include pictures of some experimental plots in the display material. Some graphs could be improved by extending y-axes only as far as data points occur.

*Reviewer #1:*

This is a well conducted experimental study in which rice monocultures are compared with rice-aquatic animal co-cultures over multiple years. Co-culture increases plant yield and adds animal yield, but requires extra input of animal feed. Overall co-culture seems to benefit yield and sustainability.

It would be interesting to discuss the broader consequences of the experimental results for application, e.g. potential costs of the additional animal feed (and additional labor?) required for the co-culture, farmer income, other ecosystem services including eutrophication and greenhouse-gas emissions.

The following are points that should be addressed in revising the manuscript:

1) Statistical analysis: it seems this was done correctly, e.g using years as subplots and not as independent measurements. However, this should be explained in more detail by presenting nominator and denominator degrees of freedom for F-tests and number of replicates in figures or legends.

2) Report amount of N (and P?) input via animal feed and arriving in soil. Discuss this in relation to other N in system and potential consequences of added P.

3) Discuss broader consequences of rice-animal co-culture in terms of economic costs/benefits to farmer, potential side effects on land used to produce animal feed and on nutrient and carbon cycle (including greenhouse-gas emissions).

4) Explain in more detail how animals were treated, grew (one or several years), affected rice plants and were "harvested" (every year all animals?).

5) It would be nice to include pictures of some experimental plots in the display material. Some graphs could be improved by extending y-axes only as far as data points occur.

*Reviewer #2:*

This manuscript reports interesting results from a rigorous set of experiments, each with six true replicates of each treatment randomized to experimental units. The authors aimed to compare rice monocultures to co-cultures of rice and an aquatic animal (carp, crab, or turtle). The data analyses appear adequate, though I do request the degrees of freedom for F-tests to double check that the repeated measures analysis was properly implemented. The findings are of widespread importance, given that rice is one of the most widely consumed crops worldwide. Increasing the yield of rice through co-culture may reduce the demand of land for rice production, leaving more land for non-human species.

I have two main suggestions for the authors, which I hope will help further improve their manuscript. First, it is necessary to clarify in visible places, including the Abstract, that feed was added in co-culture. This confounds, to some extent, the extra nutrient inputs in the feed with the comparisons between monocultures and co-cultures. It might also help to include further discussion of the extent to which the increased rice yield is likely explained by each of these two things: the animals or the nutrient inputs in the feed. The diet analyses help get at this and are another strength of this study.

Second, it would help to also briefly discuss the broader implications of these results for land use and climate change. Currently, the manuscript presents only the positive outcomes. Would changing from monocultures to co-cultures likely lead to more carbon emissions? The decomposition results suggest that this should be further investigated. Would changing from rice monocultures to co-cultures likely lead to more demand for cropland, given that soybean feed source for the aquatic animals? These implications of the results deserve a bit more attention in the Discussion section.

Title

I recommend removing 'soil nitrogen' from the title, given the non-significant results shown in Figure 1b.

Abstract

L44: here or elsewhere in the Abstract, please include a few words to let readers know that there were also feed inputs.

Introduction

L115: It would help to use a different word here, rather than 'functions' because some of the previous studies, including those in the preceding sentence, considered ecosystem functions. Also, I tend to find it more compelling when studies explain why a knowledge gap needs to be filled, rather than stating that it exists. Many things are unknown. Why should we know more about this particular knowledge gap?

L129-131: This argument seems sound if these animals do not eat the rice plants. Do these aquatic animals eat any parts of the rice plant?

Results

L147: Here and throughout, please include the numerator and denominator degrees of freedom for the F-tests. This will help readers check that the split-plot-in-time repeated measures analysis was correctly implemented.

Figure 1: Note typo at top of Figure one: carb should be crab, I believe

Figure 1: Yield should have units of mass per unit area per unit time.

The diet analyses really strengthen this study. Nicely done.

Discussion:

L288, see also Dirzo et al., 2014 Science

L294-301: Here and elsewhere, it would help to have further discussion of the implications of the results. Note that this enhanced decomposition would not only contribute to the benefit of N recycling, but would also affect the C cycle in a potentially undesirable manner.

L314-316: I am not sure that I fully understand the proposed mechanism here. Is this interpretation similar to the arguments made by McNaughton et al., (1997 Science)? It would help readers to add a bit more explanation here.

L333-334: here and in other visible places, such as the abstract, I believe it is necessary to also note that feed was added.

Materials and methods:

L338: Somewhere early on in the Methods section, please clarify how much feed was added and in what form(s).

L341-352: It would help to clarify here how each of these animal species is used, given that the production of their biomass is presented as a co-benefit alongside increased rice yield. For example, is the meat of all these species consumed by people for food?

L364-365: Do farmers typically apply pesticides? If so, then it would be interesting to include a sentence in the Discussion clarifying how the weed suppression by the animals compares, on a percentage basis, to weed suppression by pesticides.

L411: Shouldn't this percentage also be multiplied by the percentage of N in rice and aquatic animals that came from the fertilizer/feed?

L414: Please clarify how much (mass per unit area or per experimental uint) N was added as fertilizer and feed.

L443-446: This is helpful information about the feed, but more information is needed (e.g., how much N was added in the feed?) and it should be presented much earlier, as requested in comments above.

Appendix 2

L793-801: I see this helpful information here. Perhaps just move or repeat this information above, where requested in the Methods section. It will be difficult for some readers to find these important details in this appendix.

*Reviewer #3:*

Agricultural intensification has increased yields but has also negatively affected the environment. Food production needs a comprehensive overhaul in terms of sustainability, with better approaches to producing safe and healthy food, while leaving little footprint. Globally, chemical fertilizers, herbicides and pesticides are rolled back as part of our commitment to the Sustainable Development Goals (SDG) and the urgent need for a greener renaissance in sustainability (SDG 2, 12 and 13).

The coupling of two agricultural systems (i.e., integrated agriculture-aquaculture or integrated crop-livestock) have been proposed as one of the important ways that reduce the negative effects of food production on the environment: using lesser chemical fertilizers, no herbicide was needed for weed suppression, uneaten animal feed serving as nutrients for rice, no eutrophication, etc. While the idea of harnessing co-cultivation of plant-animal system for food production is great and beneficial to the environment, the availability of large-scale and long-term experimental data is lacking. Thus, this manuscript is an important contribution to the literature on integrated agriculture and aquaculture. The science in the manuscript is well thought out and executed and the manuscript is well written. The central research question – can yields in rice and aquatic animals be maintained in coupled rice-"animal" ("animal" refers to carps, crabs and turtles, collectively) systems, is timely.

The authors described their long-term and integrated agriculture-aquaculture experimental systems and the results of the multi-year study highlighted a 'solution' to the problem of historical environmental impacts caused by our current and common intensive food production systems (mainly monocultural).

The results suggested that the coupling of rice with aquatic animal production could achieve synergies between food production systems while reducing their associated environmental impacts per unit of production.

Going beyond the science, to help policy makers to better appreciate the co-cultivation strategy, a mention of the comparative financial returns (for the farmers) should perhaps be provided by the authors in their China-wide research? Such analyses (financial returns from selling rice in monocultures, selling rice and aquatic animals together) would strengthen the case for supporting various options: the crab-rice versus fish-rice co-cultivation systems in certain localities. Perhaps in certain situations (e.g. seasonality, lack of farm labour, brackish water issues), the other aquatic animals (prawns/amphibians/ducks) could be the preferred approach?

It is interesting to note that in China, rice yields were similar or higher using the rice-"animal" co-cultivation systems than under monocultures of rice. To help the readers, I would recommend that some photos for the coupled rice-"animal" ("animal" refers to carps, crabs and turtles, collectively) systems be included as a figure in this paper. The alternate is to provide some illustration to highlight the coupled rice-"animal" systems.

With the strong focus of this paper on N, which appeared to be an important (or dominant driver) factor, we still cannot rule out the plausible influence of animal manures releasing additional phosphorus (P) to drive rice growth. Thus, is there some baseline data about the dynamics in the soils and irrigation waters among the research sites? Or from past literature about P dynamics in relation to the present or absence of crabs/fishes/turtles, within a certain level of N? Current plant physiological papers, especially those involving rice growth and yield had frequently mentioned the dual roles of N and P? Some comments from the authors about P would be appropriate with reference to soil fertility and health

Other salient observations, suggestions and notes:

Importantly and across all the three co-cultivation systems, there was no requirement for weed control via herbicides. This salient finding could be included in the proposed figure,

The paper is well written and organised in a logical fashion in line with the given guidelines. If possible, some proof-reading is needed to further improve the English language.

The literature cited are relevant and updated. Appropriate statistical methods were used by the authors to analyse the data.

The sample size, field survey approaches, apparatus utilised and chemical analyses (especially stable isotopes using the IRMS, Isotope Ratio Mass Spec) were appropriate.

*Reviewer #4:*

This study by Guo et al., is set in the framework of a highly interesting system: the coculture of different species in agricultural production. In this case, not different plant species are grown together (as typically done in intercropping) but aquatic animals (carp, crabs and turtles) are cultured together with rice. Using three multi-year field experiments (one experiment for each combination between one animal species and rice plants) and three corresponding field mesocosm experiments, the authors found that generally, rice yield and apparent nitrogen-use efficiency were higher and weed infestation of the rice paddies lower in coculture with animals, compared with rice monocultures. These results are attributed to the additional feed-N unused by the animals that was shown to be taken up by plants, as well as likely feeding on insect pests and weeds by the animals and higher decomposition (also measured in the experiment) in the coculture plots. Overall, these results provide support for multispecies systems in rice production. Additionally, animal yield was measured in this study and adds to the increased economic advantage of such a system compared with rice monocultures.

This study is interesting for both, ecologists with an interest in biodiversity effects and food webs, and for agronomists aiming to increase yield while supporting agricultural diversity and limiting synthetic inputs at the same time. It is a very valuable applied study that underlines the value of diversity in agriculture. I appreciated the interesting combination of organisms across trophic levels in this culture. I also found it fascinating that this old cultural form of agriculture has a strong relevance for today's production schemes which aim for low input of synthetic fertilizers and pesticides and high organismal diversity in agricultural areas.

The experimental design of the study is clear and simple. It has relatively low power (n=6 for each treatment) but the experiments were done across several years, increasing the applicability of the results. The authors measured many relevant variables (e.g. nitrogen use efficiency) and even use C and N labelling to follow the fate of different elements in the system.

The results of the study are well supported and indicate that the coculture with animals indeed has many advantages. However, I missed a discussion of caveats and potential disadvantages of this coculture method. As far as I can see, the authors did not determine whether rice plants were destroyed or fed upon by the animals. Furthermore, coculture may pose logistical challenges such as a higher effort in management or harvest of two species that may increase costs to farmers and may make them reluctant to use this method. In addition, total nitrogen that entered the system was different between treatments (because no animal feed was used in the rice monocultures, I think), so this additional input may account for differences in yield, even if no animals are used.

I have a few major points regarding potential improvements to the analysis and presentation of the data.

Statistical analysis: I struggled to understand how the data was analysed. Was year nested within plot? Or was the experiment completely restarted every year and the data points across years can be considered independent replicates (I think they cannot, see also the temporal development in Figure 1,2)? I was also confused at times about the sample sizes behind the figures and when averages across years or replicates were used. Please be more specific here (e.g. state n in legends or on figure panels or in the text). Generally, many tests are being used where one model (testing several explanatory variables, e.g. RA/RM, time and their interaction) would likely have been the better solution.

Ethical considerations: The manuscript contains an ethics statement. However, there are few methodological details regarding the treatment of animals in the study. For example, I assume that the animals were killed in some way before drying/freezing at -20{degree sign}? How were they "harvested"?

Methodological details: I missed some details on the general method for non-experts of this culture technique. Most importantly: In this system, is harvest of the aquatic animals done every year? How long is the usual growing season? Do the animals grow as fast as rice? Even the turtles? Is the amount of fertilizer added in the experiment (550-750kg/ha) similar to what would normally be added to regular rice monocultures?

---

## [Author Response]

Essential revisions:1) Statistical analysis: it seems this was done correctly, e.g using years as subplots and not as independent measurements. However, this should be explained in more detail by presenting nominator and denominator degrees of freedom for F-tests and number of replicates in figures or legends.

Thank you very much for the comments.

(1) In our field experiments, each treatment rice monoculture (RM) or rice- animal coculture (RA) was conducted in same experimental plot every year during the 4 experimental year. We thus used years as subplots and not as independent measurements in the statistical analysis.

(2) We checked all the results of the statistical analysis. The nominator and denominator degrees of freedom for F-tests or T-tests have been provided in text. The number of replicates have been provided in legends. Please see line 151-152, line 159-169, line 177-179, line 204-209, line 221-222, line 234-243, line and Figure 1-2, Figure 4-6 of legends.

2) Report amount of N (and P?) input via animal feed and arriving in soil. Discuss this in relation to other N in system and potential consequences of added P.

Thank you very much for the comments. In the revision, the amount of N and P input *via* animal feed were applied in Appendix 2- table 1. Soil P before and after experiment in both RM and RA treatment were also provided in Appendix 1- figure 3. Because P is not our focus in this study, we did not present the data of P in the main text. But we had a brief discussion on the potential effects of feed- P on yield increase and soil P maintenance in RA compared to RM. Please see line 335-343.

3) Discuss broader consequences of rice-animal co-culture in terms of economic costs/benefits to farmer, potential side effects on land used to produce animal feed and on nutrient and carbon cycle (including greenhouse-gas emissions).

Thanks. We accepted all of these comments.

(1) Basing on this current study, our previous survey and other reported studies, we had a discussion on the differences of the economic costs/benefits between rice monoculture and rice-animal coculture, and a discussion on economic benefits that the local farmers obtain from the rice-animal cocultures. Please see line 363-375.

(2) As to the potential side effects of the rice-animal co-cultures, possible negative effects (including eutrophication, greenhouse-gas emissions) resulting from the input of feed and the increased decomposition rate were discussed in the revision. Please see line 376-395.

(3) As to the animal feed that main components are residues of soybean after the oil has been extracted. Because soybean is major oil crop in China and in other Asian countries, using residues of soybean in rice-animal coculture can achieve the recycling of resources in agriculture. Thus the integration of rice-aquatic animal coculture with soybean production would help develop sustainable rice-animal coculture. A brief discussion was provided in the revision. Please see line 396-408.

4) Explain in more detail how animals were treated, grew (one or several years), affected rice plants and were "harvested" (every year all animals?).

Thanks, we accepted all of these comments.

(1) In this study, the animals (carp, crab and softshell turtles) were released into paddy field one week after rice seedlings (15-20cm height) were transplanted. The animals (carp, crab and softshell turtles) did not affect (eat or knock down) the rice plants because the rice plants have established when the carp fry or juvenile crabs or baby turtles were released into paddy field each year.

(2) The animals (carp, crab and softshell turtles) were cultured with rice plants during the whole rice-growing period (130-150 days), and were harvested every experimental year when the rice matures.

The detail information of the animals in the experiment were provided in the Method section. Please see line 425-461, and Appendix 4. Also see Figure 7.

5) It would be nice to include pictures of some experimental plots in the display material. Some graphs could be improved by extending y-axes only as far as data points occur.

We thank this great comment. Pictures of the animals and experimental plots were provided in Figure 7.

Reviewer #1:This is a well conducted experimental study in which rice monocultures are compared with rice-aquatic animal co-cultures over multiple years. Co-culture increases plant yield and adds animal yield, but requires extra input of animal feed. Overall co-culture seems to benefit yield and sustainability.It would be interesting to discuss the broader consequences of the experimental results for application, e.g. potential costs of the additional animal feed (and additional labor?) required for the co-culture, farmer income, other ecosystem services including eutrophication and greenhouse-gas emissions.

Thanks for these valuable comments.

(1) Basing on this current study, our previous survey and other reported studies, we had a discussion on the differences of the economic costs/benefits between rice monoculture and rice-animal coculture, and a discussion on economic benefits that the local farmers obtain from the rice-animal cocultures. Please see line 363-375.

(2) Other possible effects of the rice-animal co-cultures (including eutrophication, greenhouse-gas emissions) were also discussed in the revision. Please see line 376-395.

The following are points that should be addressed in revising the manuscript:1) Statistical analysis: it seems this was done correctly, e.g using years as subplots and not as independent measurements. However, this should be explained in more detail by presenting nominator and denominator degrees of freedom for F-tests and number of replicates in figures or legends.

Thanks, we accepted all of these comments.

(1) In our field experiments, each treatment rice monoculture (RM) or rice- animal coculture (RA) was conducted in same experimental plot every year during the 4 experimental year. We thus used years as subplots and not as independent measurements in the statistical analysis.

(2) We checked all the results of the statistical analysis. The nominator and denominator degrees of freedom for F-tests or T-tests have been provided in text. The number of replicates have been provided in legends. Please see line 151-152, line 159-169, line 177-179, line 204-209, line 221-222, line 234-243, line and Figure 1-2, Figure 4-6 of legends.

2) Report amount of N (and P?) input via animal feed and arriving in soil. Discuss this in relation to other N in system and potential consequences of added P.

Thanks, we accepted all of these comments.

In the revision, the amount of N and P input *via* animal feed were applied in Appendix 2- table 1. Soil P before and after experiment in both RM and RA treatment were also provided in Appendix 1- figure 3. Because P is not our focus in this study, we did not present the data of P in the main text. But we had a brief discussion on the potential effects of feed- P on yield increase and soil P maintenance in RA compared to RM. Please see line 335-343.

3) Discuss broader consequences of rice-animal co-culture in terms of economic costs/benefits to farmer, potential side effects on land used to produce animal feed and on nutrient and carbon cycle (including greenhouse-gas emissions).

Thanks, we accepted all these comments. Combining with the comments of other reviewers, we have majorly revised the Discussion section.

(1) Basing on this current study, our previous survey and other reported studies, we had a discussion on the differences of the economic costs/benefits between rice monoculture and rice-animal coculture, and a discussion on economic benefits that the local farmers obtain from the rice-animal cocultures. Please see line 363-375.

(2) As to the potential side effects of the rice-animal co-cultures, possible negative effects (including eutrophication, greenhouse-gas emissions) resulting from the input of feed and the increased decomposition rate were discussed in the revision. Please see line 376-395.

(3) As to the animal feed that main components are residues of soybean after the oil has been extracted. Because soybean is major oil crop in China and in other Asian countries, using residues of soybean in rice-animal coculture can achieve the recycling of resources in agriculture. Thus the integration of rice-aquatic animal coculture with soybean production would help develop sustainable rice-animal coculture. A brief discussion was provided in the revision. Please see line 396-408.

4) Explain in more detail how animals were treated, grew (one or several years), affected rice plants and were "harvested" (every year all animals?).

Thanks, we accepted all of these comments.

(1) In this study, the animals (carp, crab and softshell turtles) were released into paddy field one week after rice seedlings (15-20cm height) were transplanted. The animals (carp, crab and softshell turtles) did not affect (eat or knock down) the rice plants because the rice plants have established when the carp fry or juvenile crabs or baby turtles were released into paddy field each year.

(2) The animals (carp, crab and softshell turtles) were cultured with rice plants during the whole rice-growing period (130-150 days), and were harvested every experimental year when the rice matures.

The detail information of the animals in the experiment were provided in the Method section. Please see line 425-461, and in Appendix 4. Also see Figure 7.

5) It would be nice to include pictures of some experimental plots in the display material. Some graphs could be improved by extending y-axes only as far as data points occur.

We thank this great comment. Pictures of the animals and experimental plots were provided in Figure 7. We also revised the graphs to improve the quality.

Reviewer #2:This manuscript reports interesting results from a rigorous set of experiments, each with six true replicates of each treatment randomized to experimental units. The authors aimed to compare rice monocultures to co-cultures of rice and an aquatic animal (carp, crab, or turtle). The data analyses appear adequate, though I do request the degrees of freedom for F-tests to double check that the repeated measures analysis was properly implemented. The findings are of widespread importance, given that rice is one of the most widely consumed crops worldwide. Increasing the yield of rice through co-culture may reduce the demand of land for rice production, leaving more land for non-human species.

Thank you very much for the comments. We accepted the comments about the degrees of freedom for F-tests and the repeated measures analysis.

(1) We have checked and confirmed that using "the split-plot-in-time" repeated measures analysis in our study was statistically proper because each treatment rice monoculture (RM) or rice- animal coculture (RA) was conducted in same experimental plot every year across the 4 experimental years. in our study.

(2) We checked all the results of the statistical analysis. The nominator and denominator degrees of freedom for F-tests or T-tests have been provided in text. The number of replicates have been provided in legends. Please see line 151-152, line 159-169, line 177-179, line 204-209, line 221-222, line 234-243, line and Figure 1-2, Figure 4-6 of legends.

I have two main suggestions for the authors, which I hope will help further improve their manuscript. First, it is necessary to clarify in visible places, including the Abstract, that feed was added in co-culture. This confounds, to some extent, the extra nutrient inputs in the feed with the comparisons between monocultures and co-cultures. It might also help to include further discussion of the extent to which the increased rice yield is likely explained by each of these two things: the animals or the nutrient inputs in the feed. The diet analyses help get at this and are another strength of this study.

Thanks. We accepted all these comments.

(1) The information of feed addition in rice-aquatic animal (RA) treatments were provided in abstract (line 44-45) and in Introduction (line 138-144).

(2) The amount of N and P input in the RA systems *via* feed was provided in Appendix 2- table 1. The detailed information of feed use were also provided in Method section and Appendix 4.

(3) Combining the comments of other reviewers, we have reconstructed the Discussion section. In the revision, we separately discussed the effects of animals and feed input in rice-aquatic animal coculture systems.

The aquatic animals had two important roles in these cocultured paddy ecosystems. One role involved competition, i.e., aquatic animals reduced competitors (i.e., weeds) of rice plants and thereby enhanced rice yield. A second role of aquatic animals concerned N, i.e., aquatic animals increased the recycling of N in these cocultured paddy ecosystems. Detailed discussion indicate in line 266-323.

Unlike traditional rice-fish coculture systems in which no fish feed is applied, the current coculture systems, like those described in this study, often include the application of feed in order to obtain high aquatic animal yields. The potential effects of nutrients (i.e. N and P) via feed input yield increases and soil N and P maintenance in RA compared to RM. Detailed discussion indicate in line 324-343

Second, it would help to also briefly discuss the broader implications of these results for land use and climate change. Currently, the manuscript presents only the positive outcomes. Would changing from monocultures to co-cultures likely lead to more carbon emissions? The decomposition results suggest that this should be further investigated. Would changing from rice monocultures to co-cultures likely lead to more demand for cropland, given that soybean feed source for the aquatic animals? These implications of the results deserve a bit more attention in the Discussion section.

Thank you very much for the comments. Combining with the comments of other reviewers, we have majorly revised the Discussion section.

(1) Basing on this current study, our previous survey and other reported studies, we had a discussion on the differences of the economic costs/benefits between rice monoculture and rice-animal coculture, and a discussion on economic benefits that the local farmers obtain from the rice-animal cocultures. Please see line 363-375.

(2) As to the potential side effects of the rice-animal co-cultures, it would be some possible negative effects (including eutrophication, greenhouse-gas emissions) resulting from the input of feed and the increased decomposition rate. The detailed discussion indicate in line 376-395.

(3) As to the animal feed, the main components of the feed are residues of soybean after the oil has been extracted. Because soybean is major oil crop in China and in other Asian countries, using residues of soybean in rice-animal coculture can achieve the recycling of resources in agriculture. Thus the integration of rice-aquatic animal coculture with soybean production would help develop sustainable rice-animal cocultures. A brief discussion was provided in the revision. Please see line 396-408.

TitleI recommend removing 'soil nitrogen' from the title, given the non-significant results shown in Figure 1b.

Thank you very much for the comments. Yes, no significant difference of soil N between RA and RM (Figure 1), and between before and after field experiment (Figure S1) were found. These results suggested that rice yield increased while soil N did not change in RA compared to RM.

We thus changes title to "Using aquatic animals as partners to increase yield and maintain soil nitrogen in the paddy ecosystem".

AbstractL44: here or elsewhere in the Abstract, please include a few words to let readers know that there were also feed inputs.

Thanks, we accepted this comment. The information of feed addition in rice-aquatic animal (RA) treatments were provided in abstract and main text ( Introduction, Result and Method sections). We also provided the information of the amount of N and P input in the RA systems via feed in Appendix 2- table 1. Please see line 44-45, line 138-144, line 165-166; Appendix 4.

IntroductionL115: It would help to use a different word here, rather than 'functions' because some of the previous studies, including those in the preceding sentence, considered ecosystem functions. Also, I tend to find it more compelling when studies explain why a knowledge gap needs to be filled, rather than stating that it exists. Many things are unknown. Why should we know more about this particular knowledge gap?

Thank you very much for these good comments. In the revision, we changed the sentences as "Why coculturing these aquatic animals with rice can reduce the application of fertilizers and pesticides, however, is poorly understood. Understanding how aquatic animals contribute to the reductions in fertilizer and pesticide application in coculture systems would help the development of sustainable rice production.". Please see line 114-118.

L129-131: This argument seems sound if these animals do not eat the rice plants. Do these aquatic animals eat any parts of the rice plant?

Yes, the animals (carp, crab and turtle) used in our study do not eat the rice plants.

In this study, the animals (carp, crab and softshell turtles) were released into paddy field one week after rice seedlings (15-20cm height) were transplanted. The animals (carp, crab and softshell turtles) did not eat or knock down the rice plants because the rice plants have established when the carp fry or juvenile crabs or baby turtles were released into paddy field each year. In the revision, the detail information of the animals were provided in the Method section. Please see line 425-461, and Appendix 4. Also see Figure 7.

ResultsL147: Here and throughout, please include the numerator and denominator degrees of freedom for the F-tests. This will help readers check that the split-plot-in-time repeated measures analysis was correctly implemented.

Thanks, we accepted this comment.

The nominator and denominator degrees of freedom for F-tests or T-tests have been provided in text. The number of replicates have been provided in legends. Please see line 151-152, line 159-169, line 177-179, line 204-209, line 221-222, line 234-243, line and Figure 1-2, Figure 4-6 of legends.

Figure 1: Note typo at top of Figure one: carb should be crab, I believe

Thank you very much. The correction has been done.

Figure 1: Yield should have units of mass per unit area per unit time.

Thank you very much. The correction has been done.

The diet analyses really strengthen this study. Nicely done.

Thank you very much.

DiscussionL288, see also Dirzo et al., 2014 Science

Thank you very much. We have read this nice paper by Dirzo et al., 2014 in Science and added it as a reference. Please see line 266-267.

L294-301; Here and elsewhere, it would help to have further discussion of the implications of the results. Note that this enhanced decomposition would not only contribute to the benefit of N recycling, but would also affect the C cycle in a potentially undesirable manner.

Thanks, we accepted these comments. Combining with the comments of other reviewers, we have majorly revised the Discussion section.

We have a brief discussion on the possible negative effects (including eutrophication, greenhouse-gas emissions) resulting from the input of feed and the increased decomposition rate. The detailed discussion indicate in line 376-395.

L314-316: I am not sure that I fully understand the proposed mechanism here. Is this interpretation similar to the arguments made by McNaughton et al., (1997 Science)? It would help readers to add a bit more explanation here.

Thank you very much for these great comments. Yes, the mechanism that the aquatic animals increased N-use efficiency and maintained soil N in our study was something similar to the effects of grazers in natural ecosystem as reported by McNaughton et al., (1997, Science). In the revision, we gave more explanation on this mechanism. Please see line 292-310.

L333-334: here and in other visible places, such as the abstract, I believe it is necessary to also note that feed was added.

Thanks, we accepted this comment. The information of feed addition in rice-aquatic animal (RA) treatments were provided in abstract and main text ( Introduction, Result and Method sections). We also provided the information of the amount of N and P input in the RA systems via feed in Appendix 2- table 1. Please see line 44-45, line 138-144, line 165-166; Appendix 4.

Materials and methodsL338: Somewhere early on in the Methods section, please clarify how much feed was added and in what form(s).

Thanks, we accepted this comment.

In the revision, the use of feed, and the information of amount and form of fee used in rice-carp, rice-crab and rice-turtle were provided in Introduction, Result and Method section. Please see line 44-45, line 138-144, line 165-166, line 245-261 and Appendix 2- table 1.

L341-352: It would help to clarify here how each of these animal species is used, given that the production of their biomass is presented as a co-benefit alongside increased rice yield. For example, is the meat of all these species consumed by people for food?

Thanks for these good comments. Yes, These three aquatic animals are economically important, and are usually cultured in fish ponds or paddy fields by local farmers. The meat of all these species is a popular food of the local people.

In the revision, we have rewritten this paragraph and described each of the animal species used in the coculture systems, including their economic uses, biological traits and the interactions with rice plants. Please see line 410-461.

L364-365: Do farmers typically apply pesticides? If so, then it would be interesting to include a sentence in the Discussion clarifying how the weed suppression by the animals compares, on a percentage basis, to weed suppression by pesticides.

Yes, farmers typically apply pesticides and herbicides to control insect pests and weeds in the rice monoculture. In the revision, we had a discussion on how many percentage of weeds be reduced by the animals compared to by pesticides. Please see line 283-284.

L411: Shouldn't this percentage also be multiplied by the percentage of N in rice and aquatic animals that came from the fertilizer/feed?

Yes, the total output N_y_ was obtained by multiplying the biomass of rice (grain and straw) and aquatic animals by the percentage of N in rice and aquatic animals that came from the fertilizer, feed and soil. We have provided this information in the revision. Please see line 535-538.

L414: Please clarify how much (mass per unit area or per experimental uint) N was added as fertilizer and feed.

Thanks, we accepted this comment. The information of fertilizer-N and feed-N used in each experimental plot were provided in the Appendix 2- table 1.

L443-446: This is helpful information about the feed, but more information is needed (e.g., how much N was added in the feed?) and it should be presented much earlier, as requested in comments above.

Thanks, we accepted this comment. In revision, the information of feed containing N were mentioned in the Result section and also provided in the Method section. Please see line 165-166, and line 425-461.

Appendix 2L793-801: I see this helpful information here. Perhaps just move or repeat this information above, where requested in the Methods section. It will be difficult for some readers to find these important details in this appendix.

Thanks, we accepted this comment. In the revision, the information of feed application in the Appendix were repeated in the Methods section. please see line 430-436, line 443-447, and line 455-461.

Reviewer #3:Agricultural intensification has increased yields but has also negatively affected the environment. Food production needs a comprehensive overhaul in terms of sustainability, with better approaches to producing safe and healthy food, while leaving little footprint. Globally, chemical fertilizers, herbicides and pesticides are rolled back as part of our commitment to the Sustainable Development Goals (SDG) and the urgent need for a greener renaissance in sustainability (SDG 2, 12 and 13).The coupling of two agricultural systems (i.e., integrated agriculture-aquaculture or integrated crop-livestock) have been proposed as one of the important ways that reduce the negative effects of food production on the environment: using lesser chemical fertilizers, no herbicide was needed for weed suppression, uneaten animal feed serving as nutrients for rice, no eutrophication, etc. While the idea of harnessing co-cultivation of plant-animal system for food production is great and beneficial to the environment, the availability of large-scale and long-term experimental data is lacking. Thus, this manuscript is an important contribution to the literature on integrated agriculture and aquaculture. The science in the manuscript is well thought out and executed and the manuscript is well written. The central research question – can yields in rice and aquatic animals be maintained in coupled rice-"animal" ("animal" refers to carps, crabs and turtles, collectively) systems, is timely.The authors described their long-term and integrated agriculture-aquaculture experimental systems and the results of the multi-year study highlighted a 'solution' to the problem of historical environmental impacts caused by our current and common intensive food production systems (mainly monocultural).The results suggested that the coupling of rice with aquatic animal production could achieve synergies between food production systems while reducing their associated environmental impacts per unit of production.Going beyond the science, to help policy makers to better appreciate the co-cultivation strategy, a mention of the comparative financial returns (for the farmers) should perhaps be provided by the authors in their China-wide research? Such analyses (financial returns from selling rice in monocultures, selling rice and aquatic animals together) would strengthen the case for supporting various options: the crab-rice versus fish-rice co-cultivation systems in certain localities. Perhaps in certain situations (e.g. seasonality, lack of farm labour, brackish water issues), the other aquatic animals (prawns/amphibians/ducks) could be the preferred approach?

Thanks, we accepted all these comments. Combining with the comments of other reviewers, we have reconstructed the Discussion section.-

(1) A discussion on the comparative financial returns (e.g. economic costs/benefits. product price) to farmer of the rice-animal co-cultures was provided in the revision. Please see line 356-375.

(2) We have a brief discussion on the differences of financial return among the types of cocultures (rice-carp, rice-crab and rice turtle) in the revision. We also discussed the potential application of other types of cocultures (prawns/amphibians/ducks). Please see line 396-402.

It is interesting to note that in China, rice yields were similar or higher using the rice-"animal" co-cultivation systems than under monocultures of rice. To help the readers, I would recommend that some photos for the coupled rice-"animal" ("animal" refers to carps, crabs and turtles, collectively) systems be included as a figure in this paper. The alternate is to provide some illustration to highlight the coupled rice-"animal" systems.

Thank you very much for these great comments. In the revision, we grouped the photos of the animals and experimental plots for the coupled rice-"animal" in Figure 7.

With the strong focus of this paper on N, which appeared to be an important (or dominant driver) factor, we still cannot rule out the plausible influence of animal manures releasing additional phosphorus (P) to drive rice growth. Thus, is there some baseline data about the dynamics in the soils and irrigation waters among the research sites? Or from past literature about P dynamics in relation to the present or absence of crabs/fishes/turtles, within a certain level of N? Current plant physiological papers, especially those involving rice growth and yield had frequently mentioned the dual roles of N and P? Some comments from the authors about P would be appropriate with reference to soil fertility and health

Thank you very much for these great comments.

In the revision, the amount of N and P input *via* animal feed were applied in Appendix 2- table 1. Soil P before and after experiment in both RM and RA treatment were also provided in Appendix 1- figure 3. Because P is not our focus in this study, we did not present the data of P in the main text. But we had a brief discussion on the potential effects of feed- P on yield increase and soil P maintenance in RA compared to RM. Please see line 335-343.

Other salient observations, suggestions and notes:Importantly and across all the three co-cultivation systems, there was no requirement for weed control via herbicides. This salient finding could be included in the proposed figure,

Thanks, we accepted this comment. In the revision, the information of no herbicides use in the three co-cultivation systems was provided in the legend of Figure 2.

The paper is well written and organised in a logical fashion in line with the given guidelines. If possible, some proof-reading is needed to further improve the English language.

Thanks, we have asked Dr. Bruce Jaffee who is a retire professor from University of California at Davis to help to improve the English language.

The literature cited are relevant and updated. Appropriate statistical methods were used by the authors to analyse the data.

Thank you very much for the comments.

The sample size, field survey approaches, apparatus utilised and chemical analyses (especially stable isotopes using the IRMS, Isotope Ratio Mass Spec) were appropriate.

Thank you very much for the comments.

Reviewer #4:This study by Guo et al., is set in the framework of a highly interesting system: the coculture of different species in agricultural production. In this case, not different plant species are grown together (as typically done in intercropping) but aquatic animals (carp, crabs and turtles) are cultured together with rice. Using three multi-year field experiments (one experiment for each combination between one animal species and rice plants) and three corresponding field mesocosm experiments, the authors found that generally, rice yield and apparent nitrogen-use efficiency were higher and weed infestation of the rice paddies lower in coculture with animals, compared with rice monocultures. These results are attributed to the additional feed-N unused by the animals that was shown to be taken up by plants, as well as likely feeding on insect pests and weeds by the animals and higher decomposition (also measured in the experiment) in the coculture plots. Overall, these results provide support for multispecies systems in rice production. Additionally, animal yield was measured in this study and adds to the increased economic advantage of such a system compared with rice monocultures.This study is interesting for both, ecologists with an interest in biodiversity effects and food webs, and for agronomists aiming to increase yield while supporting agricultural diversity and limiting synthetic inputs at the same time. It is a very valuable applied study that underlines the value of diversity in agriculture. I appreciated the interesting combination of organisms across trophic levels in this culture. I also found it fascinating that this old cultural form of agriculture has a strong relevance for today's production schemes which aim for low input of synthetic fertilizers and pesticides and high organismal diversity in agricultural areas.The experimental design of the study is clear and simple. It has relatively low power (n=6 for each treatment) but the experiments were done across several years, increasing the applicability of the results. The authors measured many relevant variables (e.g. nitrogen use efficiency) and even use C and N labelling to follow the fate of different elements in the system.The results of the study are well supported and indicate that the coculture with animals indeed has many advantages. However, I missed a discussion of caveats and potential disadvantages of this coculture method. As far as I can see, the authors did not determine whether rice plants were destroyed or fed upon by the animals. Furthermore, coculture may pose logistical challenges such as a higher effort in management or harvest of two species that may increase costs to farmers and may make them reluctant to use this method. In addition, total nitrogen that entered the system was different between treatments (because no animal feed was used in the rice monocultures, I think), so this additional input may account for differences in yield, even if no animals are used.

Thank you very much for the comments.

(1) We accepted the comments that the manuscript lacks a discussion of caveats and potential disadvantages of the coculture method. Combining with the similar comments of other reviewers we reconstructed the Discussion section, and had a discussion on possible negative effects (including eutrophication, greenhouse-gas emissions) resulting from the input of feed and the increased decomposition rate. Please see line 376-395.

2) As to the "---rice plants were destroyed or fed upon by the animals", the three animals (carp, crab and turtle) do not feed on rice plants because rice seedlings have grown up (about 40 days old, 15-20 cm height) when the animals are released into paddy field. The young animals do eat the hard stems of rice plants. This information has provided in Method section. Please see line 425-461.

3) As to the fee-N input, yes, rice plants used some of fee-N in the coculture compared to rice monoculture. Thus, we have a discussion on how complementary use of feed-N between rice plants and animal can contribute N use efficiency in coculture systems. We also discuss potential effect of other nutrients (e.g. P) in the feed on rice yield and soil fertility. Please see line 324-343.

I have a few major points regarding potential improvements to the analysis and presentation of the data.

Thank you very much.

Statistical analysis: I struggled to understand how the data was analysed. Was year nested within plot? Or was the experiment completely restarted every year and the data points across years can be considered independent replicates (I think they cannot, see also the temporal development in Figure 1,2)? I was also confused at times about the sample sizes behind the figures and when averages across years or replicates were used. Please be more specific here (e.g. state n in legends or on figure panels or in the text). Generally, many tests are being used where one model (testing several explanatory variables, e.g. RA/RM, time and their interaction) would likely have been the better solution.

Thank you very much for the comments.

(1) Statistical analysis: yes, the experimental year nested within plot. In our study, the same treatment (RM or RA) in each experimental plot continued across 4 years, not completely restarted every year. We thus used years as subplots and not as independent measurements in the statistical analysis.

(2) We accepted the comments that more specific explanations should be provided in legends or on figure panels or in the text. We checked all the results of the statistical analysis. The nominator and denominator degrees of freedom for F-tests or T-tests have been provided in text. The number of replicates have been provided in legends. Please see line 151-152, line 159-169, line 177-179, line 204-209, line 221-222, line 234-243, line and Figure 1-2, Figure 4-6 of legends.

(3) For the field experiment, we used a repeated measure design with four years and two culture types (RM vs RA). In our statistical analysis, we actually used repeated-measure ANOVA and focused on the effect of RM and RA, but we did not care about the effect of time or the interaction between experimental year and culture type. We thus only presented the effect of culture type in our text.

Ethical considerations: The manuscript contains an ethics statement. However, there are few methodological details regarding the treatment of animals in the study. For example, I assume that the animals were killed in some way before drying/freezing at -20{degree sign}? How were they "harvested"?

Thank you very much for the comments. In the revision, detail methods of the animal sampling and harvesting in the field, and methods of animal sample handling in lab were provided in. Method section and Appendix 7.

Methodological details: I missed some details on the general method for non-experts of this culture technique. Most importantly: In this system, is harvest of the aquatic animals done every year? How long is the usual growing season? Do the animals grow as fast as rice? Even the turtles? Is the amount of fertilizer added in the experiment (550-750kg/ha) similar to what would normally be added to regular rice monocultures?

Thank you very much for the comments.

(1) In our study, the aquatic animals were harvested every year. The growing season for rice-carp was from late May to early October (around 125days), for rice-crab was from middle May to middle October (around 150 days), for rice-turtle was from middle June to early November (around 140 days). In the revision, these information have provided in Method section. Please see line 425-461

(2) The coculture systems, both rice plants and aquatic animals grow as the time passed on. But growth rate of rice plants and aquatic animals was not the same. These information have provided in Method section.

3) Yes, to test the effects of animals, we designed the amount of fertilizer added in the experiment were similar to regular rice monocultures.